**Climate Intervention using marine cloud brightening (MCB) compared with stratospheric aerosol injection (SAI) in the UKESM1 climate model.**

Jim M. Haywood[1,2], Andy Jones[2], Anthony C. Jones[2], Paul Halloran[1], and Philip J. Rasch[3].

[1]Department of Mathematics, Faculty of Environment, Science and Economy, University of Exeter, Exeter, EX4 4QE, UK
[2]Met Office Hadley Centre, Exeter, EX1 3PB, UK
[3]Department of Atmospheric Sciences, University of Washington, Seattle WA, USA

*Correspondence to*: Jim M. Haywood (j.m.haywood@exeter.ac.uk)

**Abstract.** The difficulties in using conventional mitigation techniques to maintain global mean temperatures well below 2 °C compared with preindustrial levels have been well documented, leading to so-called 'climate intervention' or 'geoengineering'
research whereby the planetary albedo is increased to counterbalance global warming and ameliorate some impacts of climate change. In the scientific literature, the most prominent climate intervention proposal is that of stratospheric aerosol injection (SAI), although proposals for marine cloud brightening (MCB) have also received considerable attention. In this study, we design a new MCB experiment (G6MCB) for the UKESM1 Earth system model which follows the same baseline and cooling scenarios as the well-documented G6sulfur SAI scenario developed by the Geoengineering Model Intercomparison Project
(GeoMIP) and compare the results from G6MCB with those from G6sulfur. The deployment strategy used in G6MCB injects sea-salt aerosol into four cloudy areas of the eastern Pacific. This deployment strategy appears capable of delivering a radiative forcing of up to -1Wm$^{-2}$ from MCB, but at higher injection rates, much of the radiative effect in G6MCB is found to derive from the direct interaction of the injected sea-salt aerosols with solar radiation, i.e. marine sky brightening (MSB). The results show that while G6MCB can achieve its target in terms of reducing high-end global warming to moderate levels, there are
several side-effects. Some are common to SAI, including overcooling of the tropics, and residual warming of mid-and high latitudes. Other side-effects specific to the choice of the targetted MCB regions include changes in monsoon precipitation, year-round increases in precipitation over Australia and the maritime continent and increased sea-level rise around western Australia and the maritime continent; these results are all consistent with a permanent and very strong La Niña-like response being induced in G6MCB. The results emphasize that considerable attention needs to be given to oceanic feedbacks for
spatially inhomogeneous MCB radiative forcings. It should be stressed that the results are extremely dependent upon the strategy chosen for MCB deployment. As demonstrated by the development of SAI strategies which can achieve multiple temperature targets and ameliorate some of the residual impacts of climate change, much further work is required in multiple models to obtain a robust understanding of the practical scope, limitations, and pitfalls of any proposed MCB deployment.

## 1 Introduction

The difficulties in ameliorating global warming and the associated climate change via conventional mitigation are well documented (e.g., Rogelj et al., 2016; Millar et al., 2017; Tollefson, 2018; IPCC, 2018). Such difficulties have led to growing interest in so-called 'climate intervention' (also known as geoengineering) which includes proposals to deliberately brighten the planet, thereby acting to offset some of the global warming due to increased concentrations of greenhouse gases (e.g., Royal Society, 2009; Lawrence et al., 2018; Haywood and Tilmes, 2022; UNEP, 2023). Such methods for increasing the
planetary albedo are generally referred to as 'solar radiation management' (SRM). In the scientific literature, the most prominent SRM method is via stratospheric aerosol injection (SAI; e.g., Kravitz et al., 2011, 2013a, 2021; Visioni et al., 2021, 2023a), although marine cloud brightening (MCB) has also received considerable attention (e.g., Rasch et al., 2008, Jones et al., 2009, 2011; Alterskjær et al., 2012, 2013; Mahfouz et al., 2023).

Early studies of the potential impacts of MCB (e.g., Rasch et al., 2008; Jones et al., 2009) simply increased the reflectance of low-lying marine stratocumulus clouds by setting cloud droplet number concentration (CDNC) to an asymptotic maximum that was informed by aircraft observations (e.g., Martin et al., 1994; Jones et al., 2001). These early studies were subsequently improved upon by more explicit modelling through the injection of sea-salt aerosol (Jones et al., 2012; Partanen et al., 2012). However, when comparing the results from these earlier studies, difficulties became apparent in distinguishing the climatic
response in each model from the differences due to the climate intervention scenario or strategy used. Here we use 'scenario' to refer to the amount of cooling the climate intervention is intended to produce and its evolution over time, and 'strategy' for the details of the climate intervention deployment chosen to achieve the specified cooling. These difficulties contributed to the formation of the Geoengineering Model Intercomparison Project (GeoMIP; e.g., Kravitz et al., 2011, 2013a, 2015; Visioni et al., 2021, 2023b), where the primary objective was to provide standardised scenarios and strategies that could be performed
by a number of models to provide a multi-model analysis of the impacts of climate intervention proposals.

A number of studies relevant to both SAI and MCB have since been performed under the aegis of GeoMIP. The scenario most commonly used for recent GeoMIP studies of the climate impacts of SAI (experiment G6sulfur; Kravitz et al., 2015) is to reduce global mean temperature from that in a high global warming scenario to that of a more moderate one (see section 2.2
for more details). Impacts on surface climate variables (Visioni et al., 2021), stratospheric dynamics such as the North Atlantic Oscillation and Quasi-Biennial Oscillation (Jones et al., 2022), stratospheric ozone (Tilmes et al., 2022), vegetation (Xia et al., 2021) and permafrost (Liu et al., 2023) have all been assessed. The earliest GeoMIP study relevant to MCB was the G3-SSCE experiment (Alterskjaer et al., 2013) where the top-of-atmosphere radiative forcing was maintained at 2020 levels in a scenario with rising greenhouse-gas concentrations. The three participating models treated sea-salt with different degrees of complexity
ranging from fully prognostic sea-salt and CDNC, through using a climatology of sea-salt concentrations and diagnostic CDNC, to prescribed sea-salt and CDNC. Subsequently, a simpler GeoMIP experiment was defined (G4cdnc; Kravitz et al.,

2013b) where a 50% increase in the CDNC of low marine clouds was imposed over the oceans on a global basis; the simplicity of this experimental design meant that nine climate models were able to participate (Stjern et al., 2018). A more complex GeoMIP experiment called G4sea-salt (Kravitz et al., 2013b) was performed by three models that could all explicitly represent sea-salt injection into the marine boundary layer at latitudes between 30°S-30°N; this experiment highlighted that the aerosol direct effect could contribute a significant fraction of the modelled cooling (Ahlm et al., 2017).

A previous comparison of results from MCB with those from SAI (Jones et al., 2011) had a number of shortcomings. The SAI and MCB scenarios were not consistent resulting in global mean radiative forcing and temperature changes being different. The SAI simulations injected sulfur dioxide globally rather than at a specific location as the version of the model used in the study (HadGEM2; Collins et al., 2011) did not have sufficient vertical resolution or a high enough model top to allow for accurate simulation of stratospheric dynamics. Also, MCB in Jones et al. (2011) was simulated quite crudely by simply increasing CDNC in specified regions. Subsequent improvements to the treatment of MCB (Jones and Haywood, 2012) included explicit representation of injected sea-salt aerosol but the injected aerosol size distribution was assumed to be the same as that of naturally occurring sea-salt. Furthermore, all aerosols in HadGEM2 were treated as external mixtures.

In this study we present a new experiment (G6MCB) using a more up-to-date model, UKESM1. We use this experiment to examine the potential effects of MCB and compare them with those of SAI as simulated in the same model's GeoMIP G6sulfur experiment. Section 2 provides further details of UKESM1 and of the G6sulfur and G6MCB experiments. Section 3 first presents results from preliminary tests of the MCB configuration, then assesses the impact of both SAI and MCB on standard meteorological variables such as temperature, precipitation, sea-ice, and sea-level rise. Section 4 presents an analysis of whether the response to our MCB deployment strategy resembles that of La Niña. A discussion and conclusions are presented in section 5.

## 2 Model description and experimental design

### 2.1 UKESM1

UKESM1 (Sellar et al., 2019) is an Earth-system model developed jointly by the UK's Met Office and UK Universities funded under the Natural Environment Research Council and was used extensively to deliver simulations for the Coupled Model Intercomparison Project Phase 6 (CMIP6; Eyring et al., 2016). It includes an 85-level atmosphere model (Walters et al., 2019) extending to approximately 85 km altitude at a resolution of 1.25° latitude by 1.875° longitude, coupled to a 1° ocean model of 75 levels (Storkey et al., 2018). Also included are components to simulate sea ice (Ridley et al., 2018), ocean biogeochemistry (Yool et al., 2013), the land surface and vegetation (Best et al., 2011) and tropospheric and stratospheric chemistry (Archibald et al., 2020). Aerosols are represented as internal mixtures in five different log-normal modes using the GLOMAP-mode scheme (Mann et al., 2010). Aerosol components include sulfate, sea-salt, black carbon (BC) and particulate

organic matter (POM), the latter including primary and biogenic secondary POM. A variant of the Woodward (2011) bin scheme accounts for the production and transport of mineral dust (Sellar et al., 2019). The geographic distribution of the aerosol optical depth (at 550nm) for the present-day is shown in Figure 1 for reference purposes.

The activation of aerosols to form cloud droplets is described by West et al. (2014) and Mulcahy et al. (2018) and couples the dynamically evolving two-moment-modal aerosol scheme GLOMAP-mode to a Köhler-theory-based aerosol activation parameterisation (Abdul-Razzak and Ghan, 2000) to diagnose cloud droplet number concentration. Aerosol indirect effects use the PC2 (prognostic cloud fraction and condensate) cloud scheme (Wilson et al., 2008) where cloud droplet number concentration is diagnosed directly from the expected number of aerosols that are available to activate at each time step. The cloud droplet effective radius is parameterised following Martin et al. (1994) and is a function of the cloud droplet concentration, the liquid water content, cloud droplet spectral dispersion, water and air densities, and an assumed cloud base updraft velocity distribution. For further details see West et al. (2014).

## 2.2 G6sulfur

The comparison between MCB and SAI was conducted using the 'G6' framework established by Phase 6 of GeoMIP (Kravitz et al., 2015). This framework uses future scenarios developed for ScenarioMIP (O'Neill et al., 2016) and involves reducing the global mean temperature in an experiment which follows a high-emissions scenario (SSP5-8.5, experiment ssp585) to the levels in a medium-emissions scenario (SSP2-4.5, experiment ssp245) by including some form of SRM. For the G6sulfur experiment this involves injecting $SO_2$ at 18-20 km along the Greenwich meridian between 10° N and 10° S. The injection rate was modified so that, for each decade between 2021 and 2100, the decadal mean temperature in G6sulfur was within ±0.2 °C of that in ssp245. The appropriate injection rate for each decade was determined by trial and error. Three-member ensembles were used for each experiment: the three members of G6sulfur were based on three members of the ssp585 ensemble, themselves extensions of members of UKESM1's CMIP6 'historical' ensemble which in turn were initialised from different points in the pre-industrial control. Results from UKESM1's G6sulfur experiment have been documented in previous studies, e.g., Jones et al. (2021) and Visioni et al. (2021).

## 2.3. Preliminary MCB sensitivity simulations

Preliminary simulations were performed with UKESM1 to determine the optimum size bin for sea-salt injection by injecting sea-salt separately into each of bins 7-12 of the sea-salt emissions scheme (see Table 1 for the sizes of each bin). Sea-salt was injected with emission rates of 20, 50, 100, and 200 Tg yr[-1] into all four of the oceanic regions designated NP (north Pacific: 30°-50° N, 170°-240° E), NEP (north-east Pacific: 0°-30° N, 210°-250° E), SEP (south-east Pacific: 0°-30° S, 250°-290° E) and SP (south Pacific: 30°-50° S, 190°-270° E) as shown in Fig 2. Within the latitude-longitude ranges indicated, only those model grid-cells which were 100% ocean were used for sea-salt injection. By design, the areas of injection in the northern and

southern hemisphere are very similar in size. For the Northern Hemisphere, the area is 26.09 million km$^2$, while for the southern hemisphere the area is 27.25 million km$^2$. These regions were selected as they contain large areas of low-level marine cloud and are symmetrically distributed in latitude about the equator to try to avoid the detrimental effects on tropical precipitation

seen previously for hemispherically-asymmetric SAI (Haywood et al., 2013). Such detrimental results have been found to be applicable to any hemispherically asymmetric forcing mechanism that induces a significant temperature gradient across the equator (e.g. Frierson et al., 2013; Haywood et al., 2016). Previous studies using the HadGEM2 model (Jones et al., 2009; Jones and Haywood, 2012) indicated that applying MCB to clouds in the south-east Atlantic stratocumulus region could cause significant reductions in precipitation and net primary productivity over the Nordeste and Amazon regions of Brazil owing to

changes in the Walker circulation. Robust correlations have been identified between highly reflectant clouds over the south-east Atlantic, the associated localised SST reduction, and rainfall over the Nordeste region of Brazil (Hastenrath, 1990; Utide et al., 2019) and also appear to operate in UKESM1 so this region was not included in the injection strategy presented here. The preliminary simulations were performed for 15 years commencing from 2035 in the SSP2-4.5 scenario and the impact on CDNC, cloud fraction, top-of-atmosphere (ToA) net radiation and global mean temperature were assessed using data from the

last 10 years of the simulations. That the results show clear trends and tendencies suggests that analysis over this ten-year period is adequate (see results). The choice of 2035 as the start period is arbitrary and the choice of the SSP2-4.5 simulation is unlikely to impact the results as there is little deviation between SSP scenarios over this time frame.

**2.4 G6MCB**

Throughout this study, sea-salt injection was implemented by modifying the primary sea-salt emissions scheme in GLOMAP-mode which uses the Gong-Monahan approach (Gong, 2003). This is a 20-bin sectional scheme: after emission, bins 1-12 (mid-bin dry radii 1.6 nm to 0.21 µm) are mapped to GLOMAP-mode's accumulation mode, while bins 13-20 (mid-bin dry radii 0.32 to 7.0 µm) are mapped to the coarse mode. We modified emissions from a single size bin of this scheme to simulate sea-salt injection as a monodisperse spray following Salter et al. (2008) and Wood (2021); the choice of bin is described in

Section 3.1 below. The extra sea-salt is injected into the lowest model layer (layer centre at 20 m above the surface).

An experiment was set up following the GeoMIP G6 protocol (Kravitz e t al., 2015) injecting sea-salt of the optimal size as

determined from the preliminary experiments; this experiment was designated G6MCB (note that this is not an official GeoMIP-endorsed experiment, so we avoid the G6sea-salt nomenclature). Sea-salt for climate intervention was emitted concurrently and at the same rate in four ocean regions, thus effectively emissions are equal between the northern and southern hemispheres (to within 4.5%). G6MCB is also a 3-member ensemble based on the same ssp585 ensemble members as G6sulfur. As in the G6sulfur simulations, the goal of G6MCB was to reduce the global mean temperature from that of ssp585 to that of

ssp245 to within ±0.2 °C for each decade from 2021-2100, and as with G6sulfur the sea-salt injection rates for each decade were determined by trial and error.

## 3 Results

### 3.1 Selecting the optimal size bin for sea-salt injection

From Fig. 3, it is obvious that the injection of significant amounts of sea-salt into bin 7 (mid-bin radius 23 nm) is very
ineffective. The change in cloud-top CDNC is small across the range of injection rates and, along with cloud fraction, actually decreases with increasing injection rate, thereby acting counter to the objectives of MCB. These results are not dissimilar to those found for over-seeding by Alterskjær et al. (2012) and Alterskjær and Kristjánsson (2013). This reduction in cloud fraction translates to the weakest perturbation to global ToA radiative fluxes and the least global mean cooling of all the bins investigated. As the size of the injected aerosols increases through to bin 10, progressively more change in CDNC, cloud
fraction, ToA flux perturbation and global mean temperature is obtained, particularly at high injection rates, before smaller changes are seen for injections into bins 11 and 12. It therefore appears that, for UKESM1's cloud droplet activation scheme, the optimal size for aerosol injection to maximise the cooling from MCB is when the sea-salt dry radius is around 85 nm. We therefore chose injection into bin 10 for G6MCB. Some of the implications and limitations of utilizing the Abdul-Razzak and Ghan (2000) activation scheme are highlighted in section 5.

### 3.2 G6MCB compared with G6sulfur

Many of the results presented below, whether climate intervention is included or not, are compared with a nominal 'present-day' (PD); this is taken as the mean over 2015-2034 from the ssp245 experiment. Unless otherwise stated, all results are ensemble means. Figure 4(a) shows the decadal mean injection rates of climate intervention $SO_2$ and sea-salt (as dry aerosol) in G6sulfur and G6MCB, respectively. By the final decade the annual injection rate of $SO_2$ in G6sulfur (21.1 Tg yr$^{-1}$) is broadly
similar to estimates of the $SO_2$ injected by the 1991 eruption of Mt. Pinatubo (Guo et al., 2004; Dhomse et al., 2020) although of course the injection in G6sulfur is continuous rather than a pulse injection. By the same time, the sea-salt injection rate in G6MCB (413 Tg yr$^{-1}$) is a little under 10% of estimates of the observed natural global sea-salt emission rate, although the latter has a large degree of uncertainty (Lewis and Schwartz, 2004), and much of the mass of natural sea-salt emissions is in larger particles sizes not influenced by climate intervention. Figure 4(b) shows the relationships between injection rate and the
resulting decadal-mean cooling for both experiments; the data for G6sulfur are replotted with an expanded abscissa in Fig. 4(c). The two climate intervention strategies require quite different emissions to achieve a similar cooling because of differences in: 1) particle size, 2) aerosol lifetime near the surface or in the stratosphere, and 3) cloud effects. Of course, practical considerations for deployment must also be considered (i.e. the cost of deployment of SAI and MCB), but this is beyond the scope of this work. The relationship is approximately linear for $SO_2$ in G6sulfur but clearly non-linear for sea-salt
in G6MCB. The temperature-change efficiency of stratospheric $SO_2$ injection in G6sulfur is approximately constant at -126

mK / Tg [SO$_2$] yr$^{-1}$ whereas for sea-salt injection in G6MCB the efficiency falls by over a factor of three from -19.4 to -6.5 mK / Tg [sea-salt] yr$^{-1}$ as the injection rate increases over the course of the experiment (Table 2). The linearity of temperature response in G6sulfur found here may appear to run counter to the findings of Niemeier and Timmreck (2015) who found a non-linear response of radiative forcing with increasing SO$_2$ injection rates owing to the increase in particle size which decreases the scattering efficiency per unit mass at solar wavelengths, and also increases the aerosol sedimentation rate. However, they were assessing a far wider range of injection rates (0-100 Tg[SO$_2$] yr$^{-1}$) than those used in G6sulfur and the response in Niemeier and Timmreck (2015) is more linear when considered only over the more limited range of 0-20 Tg[SO$_2$] yr$^{-1}$ of G6sulfur.

Figure 5 shows an estimate of the comparative contributions to changes in ToA net shortwave (SW) radiation from cloudy and clear-sky effects in each decade of G6MCB compared with the corresponding decade in ssp245. The comparison is presented with respect to ssp245 because G6MCB and ssp245 have, by design, the same global-mean near-surface temperature through the 21$^{st}$ century; the comparison is restricted to the SW as the two experiments have very different greenhouse-gas levels. The cloudy-sky effect is estimated as the difference in SW cloud radiative effect (CRE$_{SW}$) between G6MCB and ssp245, with CRE$_{SW}$ defined as the difference between all-sky and clear-sky ToA SW fluxes:

$$CRE_{SW} = N_{SW} - N_{SW\_CS} \qquad\qquad (1)$$

Here N$_{SW}$ is the net ToA all-sky SW flux and N$_{SW\_CS}$ the same but for clear sky, and follows the convention that a negative CRE$_{SW}$ corresponds to a net loss of energy from the Earth-atmosphere system and hence a cooling effect on climate. The clear-sky effect is estimated from the difference in N$_{SW\_CS}$ between G6MCB and ssp245. By the final decade of the century, Fig. 5 shows that the sum of these estimates of cloudy and clear sky radiative effects is approximately -4 W m$^{-2}$. This is the same as the difference between the nominal forcings at 2100 of SSP5-8.5 (8.5 W m$^{-2}$) and SSP2-4.5 (4.5 W m$^{-2}$), suggesting that our method for diagnosing the components is adequate. The clear-sky effect dominates after ~2070 and is responsible for the large forcings generated by sea-salt injection towards the end of the century when the amount of cooling required to match ssp245's temperature is greatest. Although envisioned as a mechanism for cloud modification, the substantial impact of MCB on the clear sky (sometimes called 'marine sky brightening', MSB) has been found in previous studies of MCB (Jones and Haywood, 2012; Partanen et al., 2012; Muri et al., 2015; Ahlm et al., 2017).

Figure 6 shows the distribution of the cloudy- and clear-sky effects during the decades when they are at their maxima. For the cloudy-sky effect, this is 2061-2070 and Fig. 6(a) shows that the areas of greatest impact of clouds on net ToA SW correspond fairly closely with the injection regions (Fig. 2) with maxima over the sub-tropical stratocumulus regions. Even during its period of maximum impact on clouds, the change of ToA SW in G6MCB (-0.80 W m$^{-2}$) is only 0.13 W m$^{-2}$ stronger than the clear-sky effect during this same period (-0.67 W m$^{-2}$; Fig. 6c). The decade of maximum clear-sky effect on ToA SW

is 2091-2100 (Fig. 6d): the global-mean impact is -4.44 W m$^{-2}$ with regional values in the NEP and SEP injection areas in excess of -40 W m$^{-2}$. This large clear-sky effect also has to offset the fact that by 2091-2100 the global-mean cloudy-sky effect is now positive at +0.32 W m$^{-2}$ (Fig. 6c).; areas where sea-salt is injected are still areas of negative $CRE_{SW}$ changes, but dynamical feedbacks due to the large amounts of sea-salt being injected result in reductions in cloud cover and consequently positive $CRE_{SW}$ impacts in other areas. These impacts are controlled by changes in the cloud fraction that are strongly

influenced by changes in the pattern of sea-surface temperatures (SSTs; e.g. Eastmann et al., 2011) and are discussed in section 5. A warming response of clouds in simulations of MCB has also been found in earlier studies using the same cloud droplet activation scheme as UKESM1 (e.g., Alterskjær and Kristjánsson, 2013) and also in more recent studies (Mahfouz et al., 2023) that use different parameterisations (Ming et al., 2006).

Although operating at different levels of the atmosphere, G6sulfur and G6MCB both affect the climate by increasing aerosol concentrations and therefore affect aerosol optical depth (AOD). Figure 7 shows the perturbations to AOD for 2081-2100 in G6sulfur and G6MCB compared with PD: Figs. 7(a) and 7(b) show the absolute differences compared with PD while Figs. 7(c) and 7(d) show the ratio to PD. In global-mean terms the perturbation is largest for G6sulfur where AOD is more than tripled compared with the PD mean of 0.13. G6sulfur also has a more widespread distribution of geoengineering aerosol due

to the transport in the stratosphere from the injection point in the tropics and the very much longer lifetime of aerosols in the stratosphere compared with the troposphere. These changes would lead to whiter skies globally, as noted by Robock (2008). Although smaller in global-mean terms, the AOD perturbation in G6MCB is very high in the areas of sea-salt injection, especially in the tropical east Pacific with a peak local AOD of 2.4, twice the peak value in G6sulfur, reaching values that exceed present day AOD values found over continental South East Asia (e.g., Zhao et al, 2018). The AOD perturbation in

G6MCB is much more localised to the source compared with G6sulfur due to the sea-salt being injected close to the surface and the greater efficiency of aerosol removal processes in the lower troposphere which reduces the likelihood of long-range transport, especially for hygroscopic aerosol such as sea-salt.

A consequence of the greater inhomogeneity of the aerosol perturbation in G6MCB compared with G6sulfur can be seen

in Fig. 8, which shows differences between PD temperatures and the experiments. Although global-mean temperatures in G6sulfur and G6MCB follow that of ssp245, the same is not true for the latitudinal distribution of temperature. By the end of the century, Fig. 8(a) shows cooler tropics in G6sulfur and warmer polar regions compared with ssp245, with a mean pole (66.5–90° N/S)–to–tropics (23.4° S – 23.4° N) difference of 1.27 °C for 2081-2100. For G6MCB, which injects sea-salt up to latitudes of 50° N and S, the pole–to–tropics difference is increased to 1.87 °C. A discussion of the reasons for these features

for both SAI and MCB is provided in section 5.

The global distributions of the differences in near-surface air temperature between 2081-2100 and PD are shown in Fig. 9 for June-August (JJA) and Fig. 10 for December-February (DJF) for ssp585, ssp245, G6sulfur and G6MCB. The general

patterns of warming are similar in all cases (naturally more exaggerated in ssp585) with the greatest warming at high northern latitudes. However, there are some differences: there is obvious cooling over the eastern Pacific in G6MCB compared with the other experiments, as might be expected from the extremely high sea-salt AODs there and the transport patterns of the Pacific sub-tropical gyres discussed above. North America is warmer in G6MCB than G6sulfur or ssp245 in both seasons which is borne out by the probability density function of the changes which shows much wider distributions for G6MCB compared with G6sulfur (Figs. 9e and 10e). This appears to be due to the relative isolation of oceanic heat transport in the Pacific, which prevents the MCB-induced cooling from propagating more globally.

Figures 11 (JJA) and 12 (DJF) show the changes in the precipitation rate over land between the same periods as the temperature changes. For JJA, G6sulfur and G6MCB show some similarities in the patterns of precipitation change, for example the reductions in precipitation over northern and western Eurasia and parts of North America, and increased rainfall over the Sahel region in Africa and over the Indian subcontinent. However, the changes in G6MCB are more intense than in G6sulfur: e.g., the area of increased precipitation over India is more extensive, and the precipitation reduction over North America is more even than in ssp585. There are also areas where G6MCB shows quite different changes to G6sulfur, the most obvious being the increased precipitation over Australia and the pattern of changes over South America. In both cases G6sulfur shows changes very similar to ssp245 and ssp585 while G6MCB is significantly different. The situation is similar in DJF (Fig. 12) where ssp585, ssp245 and G6sulfur show broadly similar patterns of precipitation changes, while G6MCB is a clear outlier: the increased precipitation over Australia in both seasons is a noteworthy feature of G6MCB, as is the distinct increase in DJF precipitation over South America. The increase in precipitation over Australia has been diagnosed in both the GeoMIP G4cdnc (Stjern et al., 2018) and G4sea-salt (Ahlm et al., 2017) simulations with changes of the order of 10%. The simulations presented here show changes over northern Australia in JJA that exceed 500%.

The changes in annual-mean net primary productivity (NPP, i.e. the net amount of carbon produced by vegetation, diagnosed as the difference between photosynthesis and respiration) over land in 2081-2100 compared with PD are shown in Fig. 13. NPP schemes within Earth-system models generally show a strong dependence on atmospheric concentrations of carbon dioxide (the $CO_2$ fertilisation effect) and a weaker dependence on soil moisture which is a function of both precipitation and temperature: increasing precipitation increases NPP, while increasing temperature decreases NPP (e.g., O'Sullivan et al., 2020, 2022). Figure 13(a) shows a general NPP increase in ssp585 compared with PD owing to increased photosynthesis under high $CO_2$ concentrations. However, there is a significant decrease in NPP over parts of the Amazon rainforest that appears to be linked to higher temperatures and reduced precipitation (Figs. 9-12). These patterns are similar but less strong in ssp245 (Fig. 13b). NPP is higher in G6sulfur than in ssp585 owing to plant productivity not being curtailed by the high temperatures evident in ssp585, and is also higher than in ssp245 owing to the $CO_2$ fertilisation effect. The patterns of NPP change in G6MCB show rather different behaviour compared with the other experiments (Fig 13d). G6MCB shows a reduction in NPP below PD levels in the central regions of the USA, linked to the hotter and drier conditions compared with the other

experiments. G6MCB also shows significant enhancement of NPP in the tropics. In contrast to the other experiments, NPP is notably increased over Amazonia, which is the opposite effect to that found in MCB studies where the south-east Atlantic stratocumulus cloud area was targeted (Jones et al., 2009; Jones and Haywood, 2012). This indicates a strong dependence of response on the chosen injection strategy and thus a lack of generalisability of results for MCB simulations with different injection strategies, indicating that standardised emission protocols are required when reporting multi-model results. .

The change in sea-level over this period is shown in Fig. 14. All three experiments with approximately the same temperature (ssp245, G6sulfur and G6MCB) have similar amounts of global-mean sea-level rise compared with PD. G6sulfur has a fairly similar distribution of sea-level rise to ssp245, but the distribution in G6MCB is rather different, although still showing local maxima in the North Atlantic and Southern Ocean. Compared with G6sulfur and ssp245, G6MCB shows less sea-level rise in the eastern Pacific where the sea-salt injection occurs and more in the western Pacific, around the Indonesian archipelago and to the west of Australia, where the sea-level rise in G6MCB in these areas exceeds that in ssp585.

Finally, the maximum (March) and minimum (September) Arctic sea-ice areas are shown in Figure 15. Both G6sulfur and G6MCB maintain the maximum sea-ice area very close to the ssp245 levels (Fig. 15a), contrasting starkly with the area in ssp585 which diverges strongly from the others after about 2060. In contrast, there is little difference between any of the experiments for minimum sea-ice area (Fig. 15b) with all four showing an essentially ice-free Arctic in September by 2050.

## 4. How La Niña-like is the response in G6MCB?

While the patterns of near-surface air temperature (Figs. 9-10), precipitation (Figs. 11-12), and sea-level rise (Fig. 14) from the G6MCB simulations are suggestive of a La Niña-like response in the model, it is important to recognise that the results shown so far assess G6MCB for the period 2081-2100 against those of the present day. Because the objective of G6MCB (and G6sulfur) is to reduce the global mean near-surface air temperature from that of ssp585 to that from ssp245, there is inevitably some degree of global warming signal in the spatial pattern of response. In this section we examine metrics and indices such as changes in the pattern of mean sea-level pressure (MSLP) and the evolution of a simple Southern Oscillation Index (SOI). We also estimate the magnitude of internal variability and spatial patterns of La Niña response in the UKESM1 model and compare them against the difference in model response between G6MCB and ssp245 in the 2081-2100 time period, which effectively removes any global warming signal.

Taking the annual-mean MSLP between Tahiti and Darwin as a simple measure of the Southern Oscillation (Fig. 16a), neither ssp245 or ssp585 show any obvious trend, both having mean gradients of -0.02 hPa decade$^{-1}$ over 2020-2100. Note that CMIP5 simulations suggest an increase in frequency of La Niña-like conditions under global warming scenarios (Cai et al., 2015), so UKESM1 results may not be representative of the multi-model response. Over the same period the gradient in G6sulfur is -0.13 hPa decade$^{-1}$ indicating a slight tendency to more El Niño-like conditions, whereas in G6MCB the gradient is +1.02 hPa

decade$^{-1}$ indicating a marked increase in La Niña-like conditions. Figure 16 reveals that the variability in the simple Southern Oscillation Index (SOI) in UKESM1 for the SSP2-4.5, is around ±2 hPa (2 standard deviations), while the mean change in SOI by the end of the century is around +8 hPa.


Considering the spatial distribution of the change in the MSLP pressure pattern (Fig. 16b) induced by MCB under this deployment strategy, there is a strong agreement with the observed spatial patterns evident in La Niña conditions (e.g. Trenberth and Shea, 1987). To examine how much the changes in patterns of temperature and precipitation resemble La Niña, an alternate analysis is required to the patterns shown in Figs. 9-12 as they are a composite of responses to both MCB
deployment and to global warming. To isolate the response in the absence of global warming, we analyse G6MCB – ssp245. We also analyse the strongest five La Niña-like events from a century long pre-industrial simulation which has negligible temperature trend. The five strongest La Niña-like events are determined as those years with the strongest positive SOI and a mean is calculated from those five years for both temperature and precipitation. The perturbation in temperature and precipitation is than calculated as the different between the mean of these five years from the mean from the 100year
simulation. The patterns of temperature change and precipitation change are presented in Figs. 17 and 18.

Fig. 17 shows that the spatial pattern of natural variability in near-surface air temperatures in UKESM1 over the Pacific shows many similarities to that diagnosed in earlier versions of UKESM1 (e.g. Collins, 2005) with a maximum closely confined to the equator while observations suggest a broader maximum. As expected from the temporal analysis of SOI (Figure 16a), the
magnitude of the perturbations in spatial distribution of temperature in the MCB scenario far exceed those from natural variability. The spatial patterns of the temperature change from MCB bears many similarities to the patterns diagnosed from natural variability, particularly during the DJF season. In DJF, regions where the spatial patterns are similar include the cooling over the east Pacific, and the strong warming impact over the USA, and the strong cooling over Alaska, and the cooling over Australia. However, there are significant differences in the near surface air temperature response over some other areas such
as central and eastern Europe and S.E. Asia, but the general pattern strongly suggests La Niña-like climate change.

Fig. 18 shows that, again, the magnitude of the precipitation response is greater in the MCB simulations than in the natural variability. In DJF, again the agreement in precipitation pattern between the MCB perturbation simulations and natural variability shows some coherence, with a strong increase in precipitation over Australia, a similar pattern across South
America, and drying across the Atlantic from Florida to northern Europe. In JJA there is evidence of increased precipitation over the Maritime continent, the Indian sub-continent and northernmost south America in both the MCB simulations and model natural variability.

The analysis of the MCB induced changes in the pattern and magnitude of the MSLP and the patterns of the near-surface air temperature and precipitation lead us to conclude that the response is La Niña-like for this specific MCB deployment strategy.

## 5. Discussion and Conclusions

The objective of the simulations presented in this study was to reduce global mean temperatures from those of the SSP5-8.5 scenario to those of SSP2-4.5 using SAI (G6sulfur) and MCB (G6MCB). Such simulations have been performed by multiple

models for the G6sulfur experiment (e.g., Visioni et al., 2021; Jones et al., 2022; Tilmes et al., 2022). These simulations generally show that such an approach reduces many detrimental impacts associated with climate change in SSP5-85 such as global and regional temperatures and high-latitude precipitation (Visioni et al., 2022), permafrost loss (Liu et al., 2023), or changes in sub-tropical atmospheric river activity (Liang and Haywood, 2023). However, there remain significant residual impacts on stratospheric dynamics and ozone (Jones et al., 2022; Tilmes et al., 2022) and on climate impacts at the surface

such as a general reduction in global precipitation, particularly in mid-latitude and tropical areas (Visioni et al., 2021) and increased drought over southern Europe (Jones et al., 2022). It is also thought that high aerosol concentrations from $SO_2$ injections into the lower stratosphere, in its non-neutralised form of sulfuric acid, could cause long-term issues for aircraft engines, airframes and other aviation components such as windows (e.g., Schmidt et al., 2014, and references therein), significantly reducing their servicing intervals and increased the associated operating costs.


The latitudinal distribution of aerosol optical depth in G6sulfur peaks in tropical regions which is due to the specified injection strategy of injecting between 10° N and 10° S. Significant work has been done examining the utility of alternative strategies using latitudinally variable injections (e.g., Kravitz et al., 2017; Bednarz et al., 2023; Visioni et al., 2023a; Henry et al., 2023) that reduce the tropical AOD peak and the associated over-cooling of tropical regions with continued warming at

high latitudes (Fig. 8c). The magnitude of the peak in AOD for equatorial injections is also affected by the model-dependent strength of the tropical pipe which acts as a barrier to equator-to-pole transport. Compared with UKESM1, the CESM2 model for example displays less confinement of sulfate aerosol to the tropics for equatorial injections (Jones et al., 2021).

The G6MCB simulations presented here also deliver the primary objective of the climate intervention scenario. The strategy

for achieving this is by targeting those areas where clouds are considered to be most susceptible to aerosol injection (e.g., Latham et al., 2008), as shown in Fig. 2. It was found that the optimal size for injection of sea-salt aerosols in UKESM1 was around 85 nm radius, considerably larger than that suggested by process-level modelling studies (e.g., Connolly et al., 2014; Wood, 2021), although this may be an artifact of the choice of aerosol activation parameterization as discussed below. The aerosol indirect effect (aerosol-cloud interaction) was found to saturate i.e., suffer significantly from diminishing returns,

becoming secondary to the cooling impact of the aerosol direct effect (aerosol-radiation interaction), an effect which has been noted before (e.g., Ahlm et al., 2017). At sufficiently large injection rates, the forcing from aerosol-cloud interactions was

found to swap sign from negative to positive (see also Alterskjær and Kristjánsson, 2013, and Mahfouz et al., 2023). Alterskjær and Kristjánsson (2013) suggest that deliberate injections into the nucleation mode can lead to a significant positive forcing (warming effect), because of the strong competition for water vapour between a large number of small sea-salt particles. This leads to many hydrated aerosols, but a reduction in the relative humidity and a reduction in the cloud fraction. The injection of coarse mode particles (Alterskjær and Kristjánsson, 2013) and over-seeding of accumulation mode aerosols in areas of high background aerosol concentrations (Alterskjær et al., 2012) have also been found to exert a significant positive forcing due to a decrease in the activation of background aerosols. These results contrast with those of Wood (2021) who used a heuristic model and large eddy simulations to suggest a maximum radiative forcing efficiency for much smaller aerosols in the range 15-30 nm radius (i.e. in the Aitken mode). Wood (2021) also notes that the results may be specific to climate models that utilise the parameterization of Abdul-Razzak and Ghan (2000; hereafter ARG) for aerosol activation and the positive radiative forcings reported by Alterskjær and Kristjánsson (2013) may be an artefact of the scheme's incorrect representation for water vapour competition at very high concentrations of small particles. Limitations of the ARG activation scheme are also highlighted by Ming et al. (2006) and by Nenes and Seinfeld (2003) who suggest that the scheme does not perform well for marine aerosol owing to biases introduced by empirical correlation.

In our study, while the microphysical impacts of clouds are evident at more modest injection rates (Fig 6a), the dynamical response of clouds becomes increasingly important as the injection rates increase (Fig 6b). Robust observational correlations between cloud fraction and SSTs have been developed on a regional basis from observations (e.g. Warren et al., 2007; Eastman et al., 2011) which reveal strong negative correlations between SSTs and clouds (i.e colder SSTs lead to more clouds) in regions of upwelling over the eastern pacific, which transition to strong positive correlations (i.e. colder SSTs lead to less clouds) in the central Pacific. In our simulations, the strong local cooling that is induced over the eastern Pacific by the MCB is advected equatorward and then westward, leading to an SST-related reduction in cloud fraction over the central and western Pacific. These model results are therefore in line with observations that relate SSTs to cloud fraction (Eastman et al., 2011) and also with observations of the response of clouds to La Niña-like conditions (Park and Leovy, 2004) which are discussed in more detail later.

On the face of it, it might be concluded that MCB may be viable in delivering relatively modest radiative forcings of up to around -1Wm$^{-2}$ for this particular injection strategy, but radiative forcings stronger than around -1Wm$^{-2}$ may not be achievable through MCB. An alternative interpretation may be that the ARG scheme may produce reasonable results when the injection rates of sea-salt are low, but that it becomes progressively less reasonable when the injection rates become very high. Thus, the swap-over seen in G6MCB from the cooling being dominated by aerosol-cloud interactions to being dominated by aerosol-radiation interactions may be an artifact of pushing the ARG activation scheme beyond the conditions that it was designed for. Work is ongoing to examine whether other activation schemes such as those based on Nenes and Seinfeld (2003) might produce significantly different results.

In G6MCB, the distribution of aerosol optical depths shown in Figure 7 suggests a semi-permanent MCB-induced 'hydrated-aerosol fog' over the injection regions by 2100, particularly over the NEP and SEP regions. In these areas the AOD at 550nm reaches values of around 2, which would mean that even in cloud-free conditions, less than 2% of direct solar radiation would reach the surface of the Earth for a mean solar zenith angle of 60°. Impacts of changes in the diffuse/direct fraction of sunlight have been investigated for terrestrial ecosystems (e.g., Mercado et al., 2009) but less attention has been given to any potential impacts on marine ecosystems (e.g., Morel, 1991). Using an empirical relationship between the surface layer aerosol extinction coefficient and visibility (Koschmeider, 1924) suggests that, averaged over the injection regions, the annual mean atmospheric visibility is reduced to approximately 6 km. Whether such a permanent fog over the eastern Pacific could cause a hazard to shipping is beyond the scope of this study.

Multi-model GeoMIP studies have documented that reducing the solar constant by a fixed fraction reduces downward shortwave flux by a greater amount in the tropics than at the poles and will have no impact at all in wintertime for polar regions where there is no solar irradiance (Kravitz et al., 2013). In addition, the fact that UKESM1 exhibits a strong tropical pipe that isolates the tropical stratosphere from the mid-latitudes inhibits poleward transport of aerosols, resulting in an aerosol optical depth that is much greater in tropical regions than over the poles (e.g. Figure 7a and Visioni et al., 2023). Thus, G6sulfur shows the expected maximum zonal mean residual warming for 2081-2100 between 60-90 °N which has been evident in GeoMIP simulations which inject aerosol at Equatorial latitudes (e.g., Kravitz et al., 2013a, 2015). ,

For MCB, in the northern hemisphere, much of the cooling impact from MCB is confined to the low-latitude and eastern Pacific, accompanied by warming in the Kuroshio and North Pacific Current region (Figs 9, 10, 17). This Pacific Decadal Oscillation (PDO)-like pattern of SST change, like the PDO itself (Newman et al., 2016), appears to arise from a combination of multiple oceanographic and atmospheric processes. Enhancement of the high pressure systems sitting above the subtropical North and South Pacific in response to MCB (Fig 16b) will impact the ocean in a number of ways. (1) Increased equatorward windspeeds along the west coasts of North and South America, will increase Ekman transport and upwelling of cool water along those coasts, supressing SSTs towards the east of the basin. (2) Increased anticyclonic movement of air above the subtropical gyres will result in increased geostrophic flow within the gyres, evidenced by positive sea surface height anomalies over the gyres (Fig 14d). With a strengthening of the subtropical gyre circulation there will be an increase in southward then westward transport of cool waters on the eastern side of the basin, and an increased northward transport of warm water in the western side of the basin. (3) Strengthening of the subtropical gyres will result in increased Sverdrup transport equatorward across the gyres, balanced by an enhancement of the western boundary currents (Vallis et al., 2017), in the case of the North Pacific, the Kuroshio current. Strengthening of the Kuroshio current will transport more warm equatorial water, more quickly, to the inter-gyre boundary region, where the secondary maximum in SSTs is seen (Figs. 8, 17). Similar arguments can be made for the strengthening of the South Pacific sub-tropical gyre. Thus, while the overcooling in the tropics in SAI simulations is

linked to changes in the surface irradiance, for MCB the overcooling in tropical regions in this study appears to be influenced by the ocean circulation. We note that, for SAI where considerable research has been performed into strategies to ameliorate residual temperature impacts by injecting at latitudes outside of the tropics (e.g., Kravitz et al., 2017; Henry et al., 2023), residual temperature impacts from MCB will be an even stronger function of the deployment strategy owing to the inhomogeneous nature of any deployment.

The fact that sea-level rise in areas such as western Australia and the maritime continent is more significant in G6MCB than in the baseline high-end global warming SSP5-8.5 scenario is a notable feature. This, and many of the features evident in the seasonal changes in precipitation, appears to be associated with the deployment strategy used in G6MCB inducing a La Niña-like response. Given that G6MCB targets regions of low cloud associated with the upwelling of cold water off the western coasts of the North and South American continents, it is not surprising that the cooling pattern over the Pacific resembles that of La Niña. There is clear observational evidence from tide gauge and satellite altimetry data of enhanced sea-levels along the entirety of the western and northern Australian coasts during La Niña conditions (McInnes et al., 2016) while the opposite occurs during El Niño (e.g., Nerem et al., 2009; Widlansky et al., 2017). While the physical attribution of erosion of coastlines is complicated by the impacts of storm frequency and intensity and of rainfall, enhanced erosion has been attributed to La Niña in areas of the west Pacific including the north and west Australian coastlines (e.g., Vos et al., 2023). Sea-level variations of as much as +20-30cm have been observed over low lying islands of the Western Pacific during La Niña conditions and with similar magnitude negative anomalies during El Niño conditions (Becker et al., 2012). Given the vulnerability of these islands to sea-level rise, and the fact that the La Niña-like patterns induced in our model simulations are many times greater than model natural variability as evident from the trends in SOI, these implications clearly motivate additional study and further exploration of MCB emission scenario choices.

In addition to the impacts of sea-level rise, La Niña is associated with increased precipitation over Australia, the maritime continent, north-eastern South America, the north of the Indian subcontinent and the Sahel region of Africa during JJA; La Niña is also associated with decreased precipitation over central and southern USA and southern areas of south America (e.g., Ropelewski and Halpert, 1989; Mason and Goddard, 2001). These patterns are all evident in the G6MCB simulations. Because the changes in the SOI are so much stronger than those of natural variability (Figure 16), it is possible that such changes could lead to large-scale marine ecosystem collapse. Impacts of global warming on the productivity of regional fisheries are underway (Fish-MIP, Tittensor et al., 2018) but it would be prudent to examine impacts under any proposed future MCB strategies. While there has been much debate as to whether the cooling due to stratospheric aerosols from explosive volcanic eruptions induces an El Niño type of response, the analyses reveal no generalisable conclusions (Self et al., 1997; McGregor et al., 2020) and there is little evidence of a general El Niño-like induced response in G6sulfur.

A trend in the future mean climate into La Niña-like conditions would have profound impacts on regional climate, with implications for climate resilience and adaptation. On a global basis, fish provides around 11% of human protein consumption (FAO, 2014). While Peruvian fisheries generally report increased yields under La Niña conditions in the observational record (e.g. Bertrand et al., 2020), the La Niña-like conditions induced under this specific scenario and strategy are many times stronger than those that occur due to natural variability.

It needs to be emphasised that the MCB results presented here are strongly dependent on both the scenario (the amount of cooling required) and the deployment strategy (the regions where sea-salt is injected) being considered. The areas chosen for sea-salt injection here are simply plausible, i.e. they have large amounts of low-level cloud that are susceptible to cloud-seeding. This is just one choice from any number of injection distributions which could be defined, especially in the absence of any real-world constraints because of the purely theoretical nature of large-scale marine cloud brightening technology. The results from G6MCB are therefore specific to this choice of injection strategy. The results suggest that MCB may indeed be relatively effective for this scenario and strategy during initial deployment: for example, Fig. 4(b) suggests a global-mean cooling of around 1.5 °C for an injection rate of ~100 Tg yr$^{-1}$ of sea-salt. However, this cooling efficiency falls (Table 2) as areas that were initially susceptible to modification become progressively less susceptible as injection rates increase, and the direct aerosol radiative effect starts to dominate. However, the clear evidence for a La Niña-like climate response produced by this and similar injection strategies (more cooling in the eastern compared to the western pacific, also found in Jones and Haywood (2009), Rasch et al (2009) and Hill and Ming (2012)) clearly needs to be considered. Designing a more nuanced strategy should be the focus of more research. The very inhomogeneous forcing of MCB as applied in this scenario, appears to induce specific changes in the oceanic circulation in the Pacific sub-tropical gyres that transport the MCB-induced SST perturbations equator-wards and westwards. While SAI has been examined for the most part by atmospheric scientists, for MCB it appears essential to include more detailed analyses by oceanographers to better understand and quantify any potential impacts. Note also that MCB will be susceptible to the termination effect, if climate intervention is stopped abruptly (e.g., Jones et al., 2013, MacMartin et al., 2022) due to the short lifetime of MCB aerosols in the troposphere.

Despite the difficulty of generalising with regard to MCB, some factors are nevertheless likely to remain constant. For a given global-mean forcing, MCB will be characterised by smaller regions of high forcing compared with the larger (or global) areas of lower forcing characteristic of SAI, i.e. the forcing from MCB is always likely to be more inhomogeneous than that from SAI. There is always likely to be some ambiguity between MCB per se (effects on clouds) and MSB (direct aerosol radiative effects). It is therefore important that modelling studies should specifically simulate the injected sea-salt aerosol and not just modify CDNC values as was done in early investigations. A caveat with all studies reporting results from aerosol-cloud interactions within a coarse resolution Earth System Model, is that many of the microphysical processes such as cloud top cooling, subsidence, entrainment, detrainment, the representation of cloud base-updraft velocities etc. are not explicitly resolved or represented (e.g. Stevens and Feingold, 2009; Seifert et al., 2015; Haghighatnasab et al., 2022) which contributes

to a significant uncertainty in results of global MCB studies. Large-scale effusive volcanic eruptions provide useful, but not perfect analogues for examining the representation of MCB within such coarse resolution models. The results from these studies reveal reasonable representation of the aerosol-induced observed perturbations to cloud droplet effective radius within coarse resolution climate models (e.g. Malavelle et al., 2017; Chen et al., 2022), but shortcomings in the representation of aerosol-induced perturbations to cloud fraction (e.g. Chen et al., 2022).

Clearly much more research is needed if the complexities of aerosol-cloud-interactions and the associated coupling of the ocean and atmospheric circulations are to be fully understood and if MCB strategies are to be represented with fidelity in future climate scenarios.

**Code and data availability**

UKESM1 model data for the ssp585, ssp245 and G6sulfur experiments are available from the Earth System Grid Federation (WCRP, 2021). Data from G6MCB are available on request from the authors.

**Author contributions**

JMH and AJ devised the G6MCB experiment with input from PJR, performed the analysis and wrote the paper. AJ performed the G6sulfur and G6MCB simulations. ACJ and PJR provided comments on earlier versions of the manuscript. PH provided an assessment of the influence of the MCB deployment on the oceanic circulation.

**Financial support**

JMH, AJ, and PJR were supported by SilverLining through its Safe Climate Research Initiative. JMH and AJ were also supported by the Met Office Hadley Centre Climate Programme funded by DSIT. JMH was further supported by the NERC EXTEND project (NE/W003880/1).

**Competing interests**

The authors declare that they have no competing interests.

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

**Tables**

955                       **Table 1.** The sea-salt emission scheme bin sizes tested for G6MCB (nm).

| Bin number | Mid-bin dry radius (nm) |
|---|---|
| 7 | 22 |
| 8 | 36 |
| 9 | 55 |
| 10 | 86 |
| 11 | 133 |
| 12 | 207 |

**Table 2.** The average efficiency of sea-salt injection in changing global-mean near-surface temperature as a function of the rate of sea-salt injection in G6MCB.

| Injection rate (Tg yr$^{-1}$) | Efficiency (mK Tg$^{-1}$ yr) |
|---|---|
| <100 | -19.4 |
| 100 – 200 | -12.3 |
| 200 – 300 | -8.5 |
| 300 – 400 | -7.3 |
| >400 | -6.5 |


**Figures**

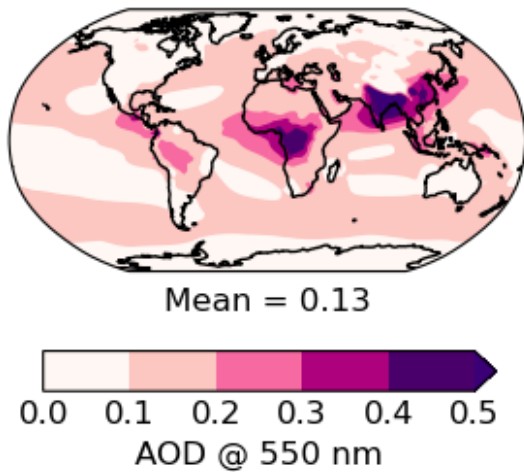

**Figure 1.** Showing the global annual mean aerosol optical depth (AOD) diagnosed for present day conditions for UKESM1.

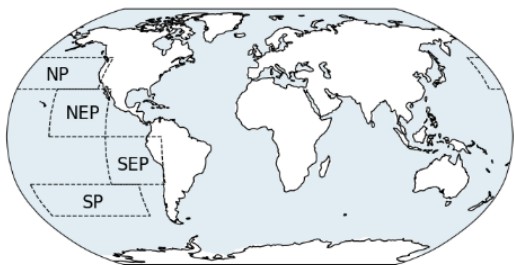

970          **Figure 2:** The regions used for sea-salt injection in G6MCB; only ocean points within each region were used.

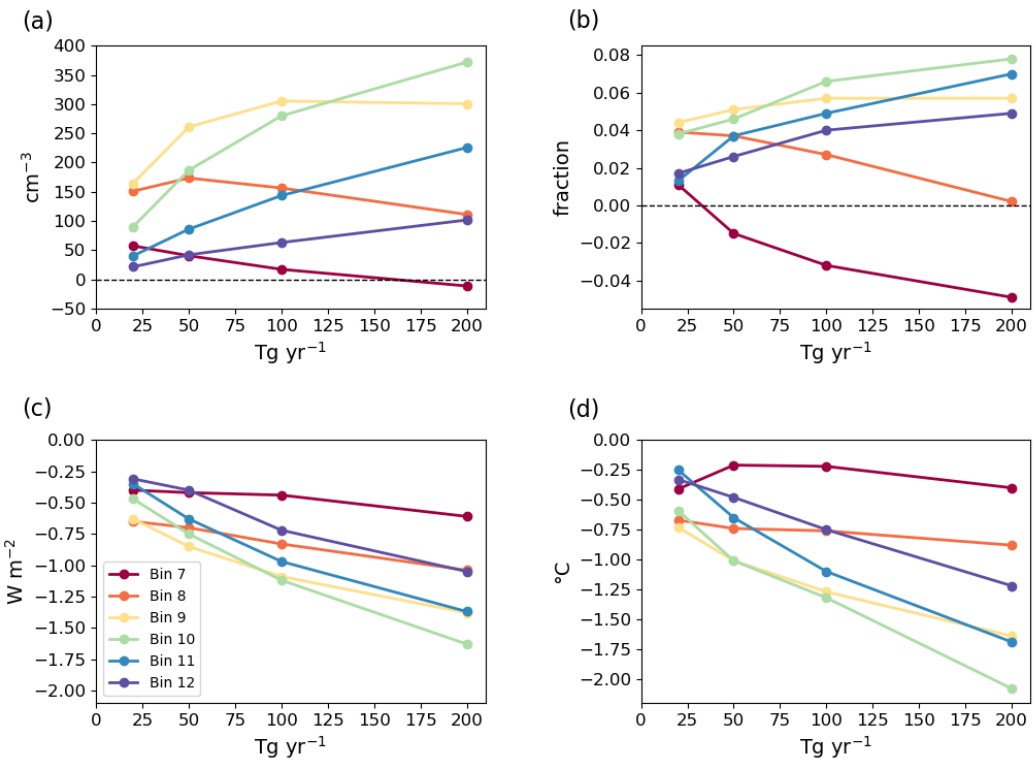

**Figure 3:** 10-year mean changes with respect to a non-perturbed control as a function of sea-salt injection rate in UKESM1 simulations using different sea-salt emission size bins: (a) cloud-top CDNC averaged over the four injection regions ($cm^{-3}$), (b) cloud fraction averaged over the four injection regions, (c) global-mean ToA net radiation ($W\ m^{-2}$), (d) global-mean near-surface air temperature (°C). The sizes of bins 7-12 are given in Table 1.

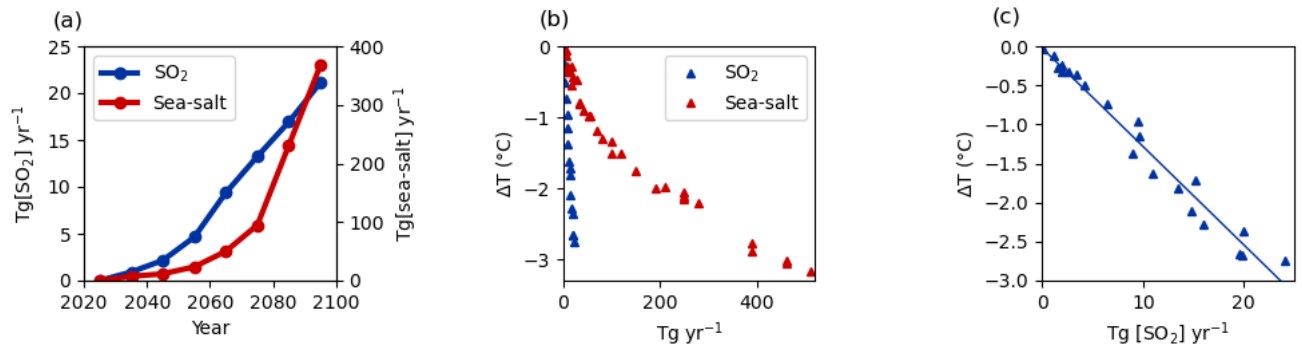

**Figure 4:** (a) Ensemble mean decadal injection rates of $SO_2$ and dry sea-salt mass in G6sulfur and G6MCB (Tg yr$^{-1}$); note the different scales. (b) Decadal-mean temperature changes due to $SO_2$ and sea-salt injections as a function of injection rate

(°C). (c) The same as (b) but rescaled to only show $SO_2$ with a least-squares straight line fit added. Panels (b) and (c) show

data from individual G6sulfur and G6MCB ensemble members. Panel (b) also includes G6MCB data from attempts which

did not meet the G6 protocol's temperature criterion (i.e. maintaining the decadal global mean temperature within ±0.2C of

that of ssp245) but are included as they are still indicative of the relation between sea-salt injection rate and temperature

change.


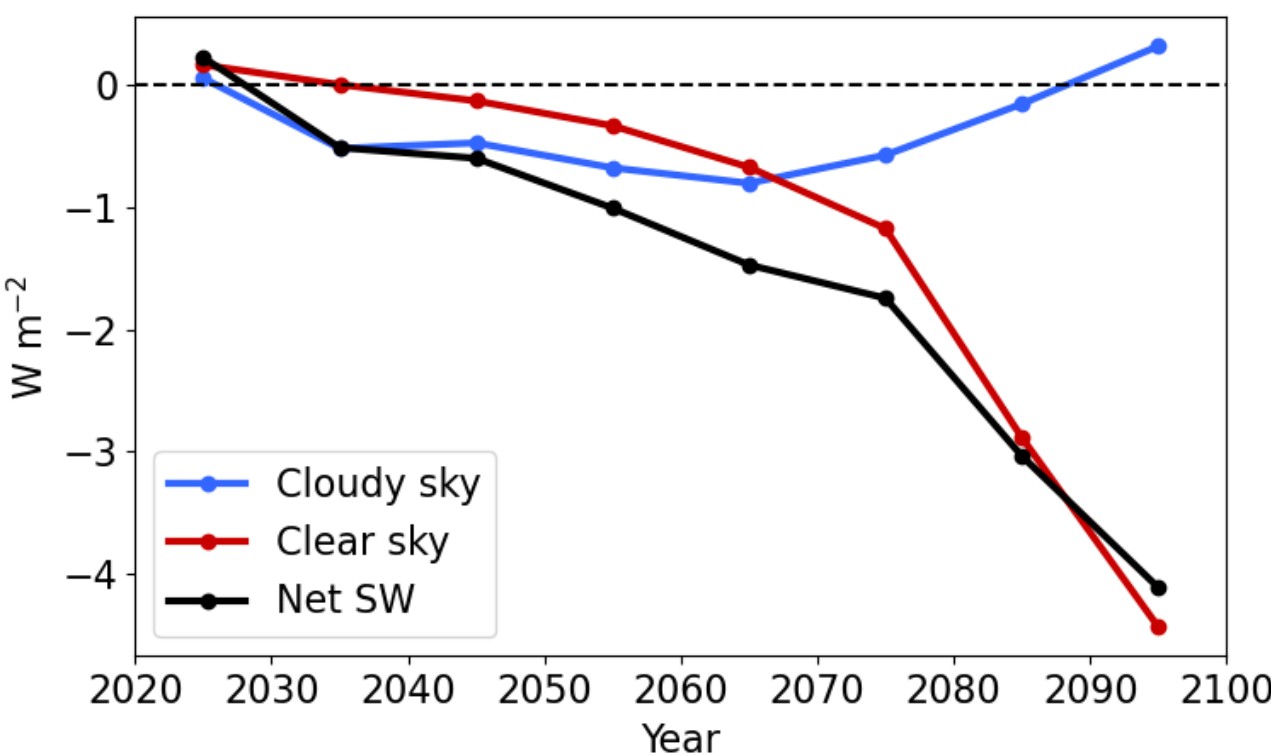


**Figure 5:** Ensemble-mean estimates of the cloudy-sky, clear-sky, and net solar contributions to the difference in decadal-mean ToA net downwards SW radiation between G6MCB and ssp245 (W m$^{-2}$).

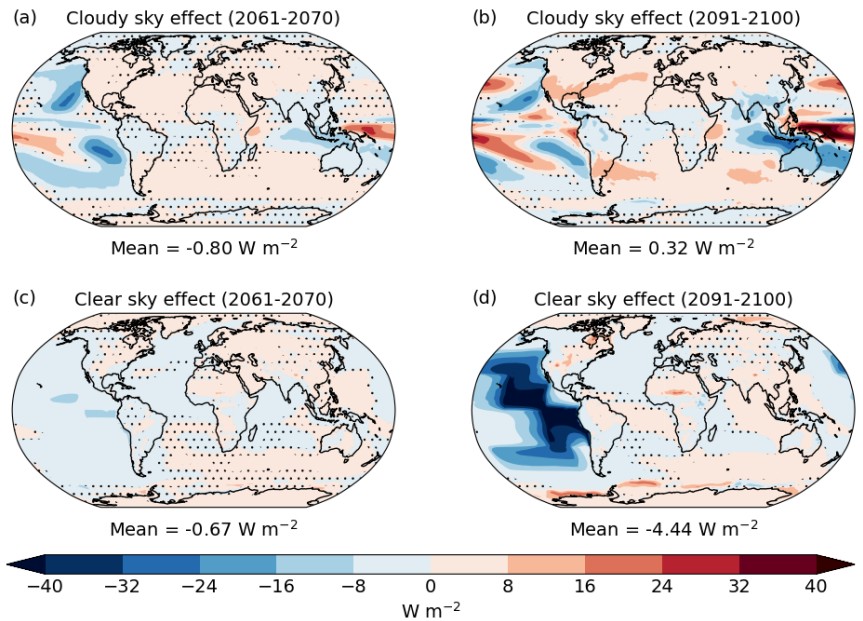


**Figure 6:** The decades of maximum contribution from the cloudy- and clear-sky effects of MCB in terms of ToA net SW (G6MCB minus ssp245; W m$^{-2}$): 2061-2070 is the decade of maximum cloudy-sky effect (panels (a) and (c): left column) and 2091-2100 the maximum for the clear-sky effect (panels (b) and (d): right column). Stippled areas show where the differences are not significant at the 5% level in a two-tailed *t*-test.


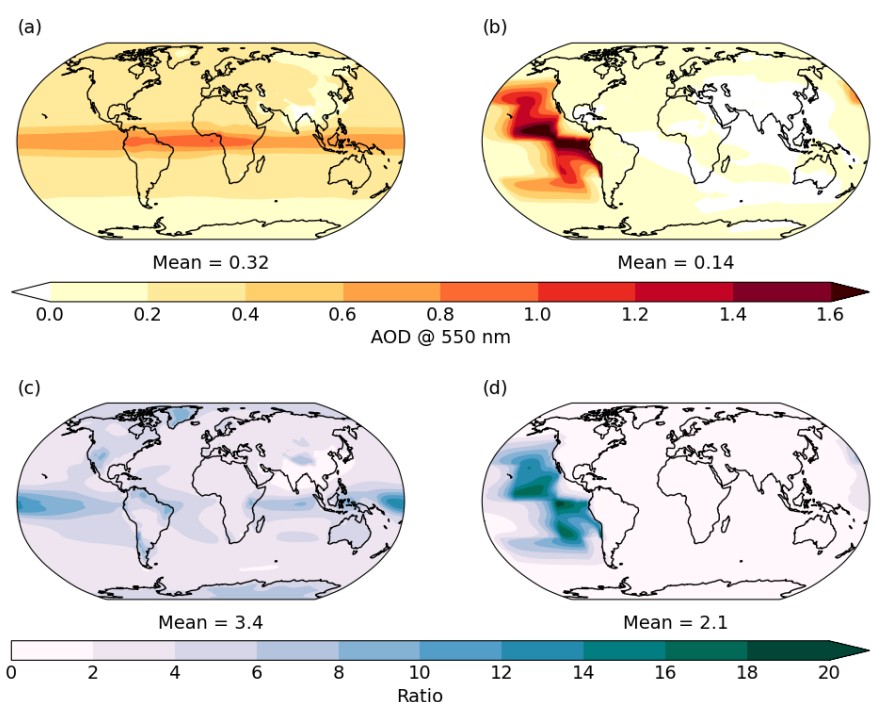


**Figure 7:** (a) The difference in AOD at 550 nm for 2081-2100 in G6sulfur compared with present-day. (b) Same as (a) but for G6MCB. (c) The ratio of AOD between G6sulfur (2081-2100) and PD. (d) Same as (c) but for G6MCB.


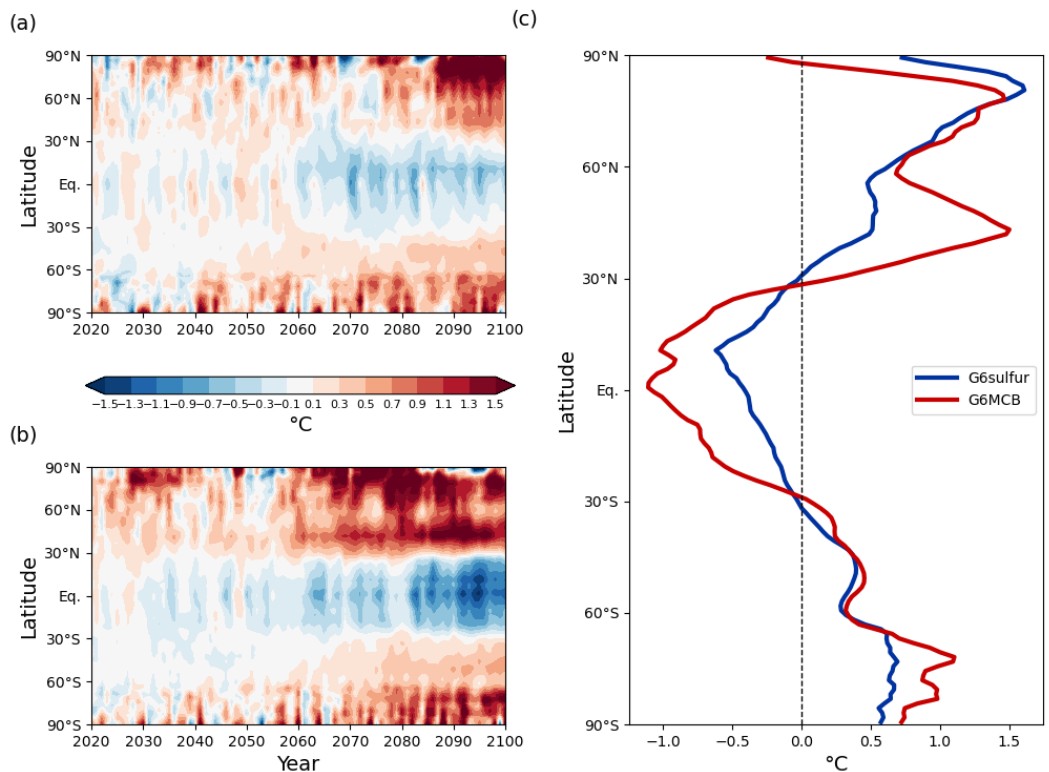

**Figure 8:** (a) Time-latitude evolution of the difference in near-surface air temperature (°C) between G6sulfur and ssp245. (b) The same as (a) but for the difference between G6MCB and ssp245. (c) Zonal means of the temperature differences for 2081-2100.


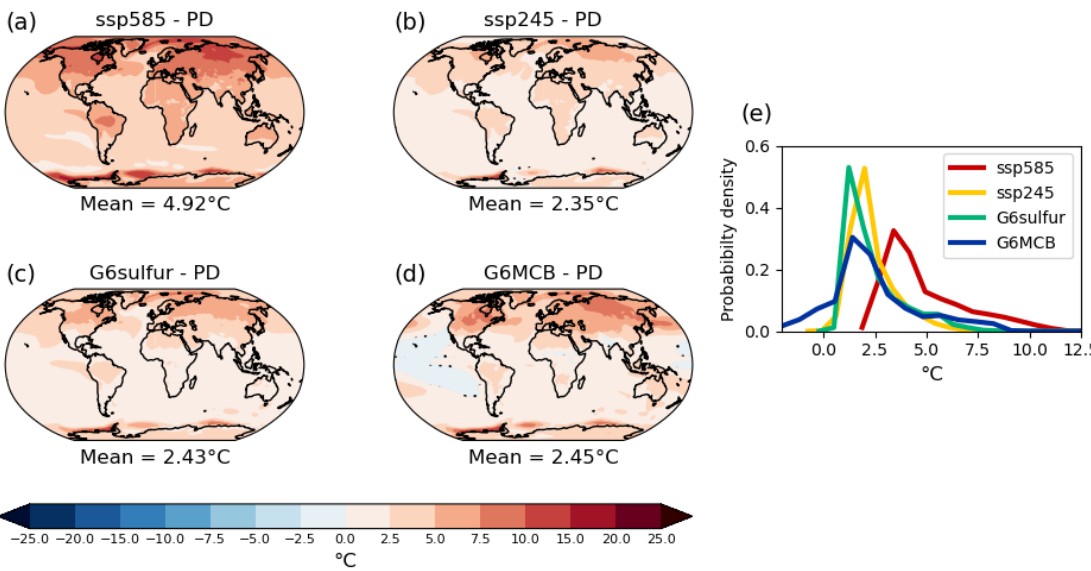


**Figure 9:** Change in JJA near-surface air temperature (°C) for 2081-2100 compared with PD in (a) ssp585, (b) ssp245, (c) G6sulfur and (d) G6MCB. Stippled areas show where the differences are not significant at the 5% level in a two-tailed *t*-test. (e) Probability density function of the changes.


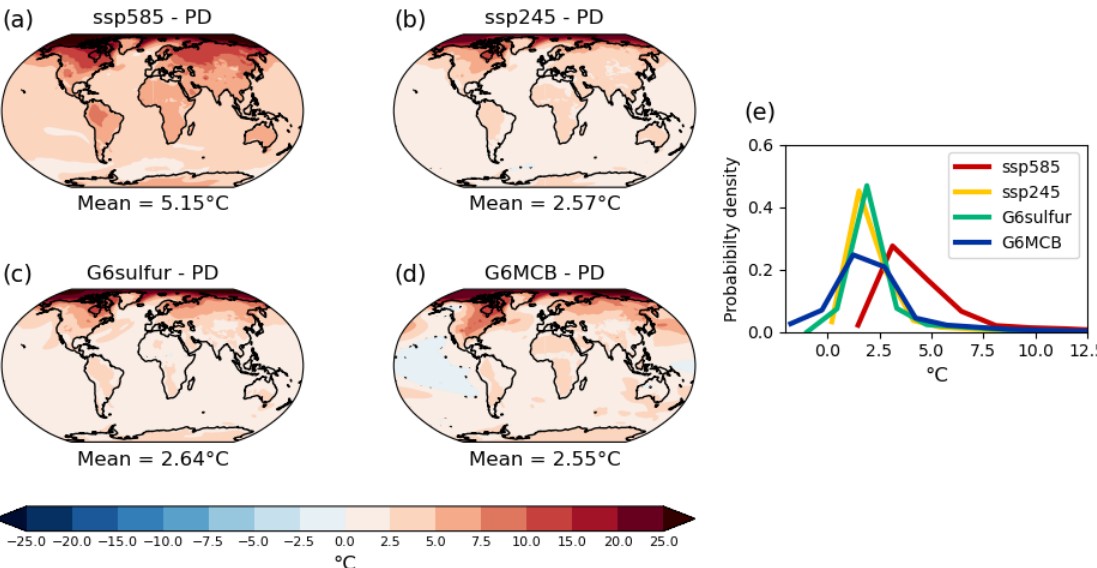

1035            **Figure 10:** Same as Fig. 9 but for DJF.


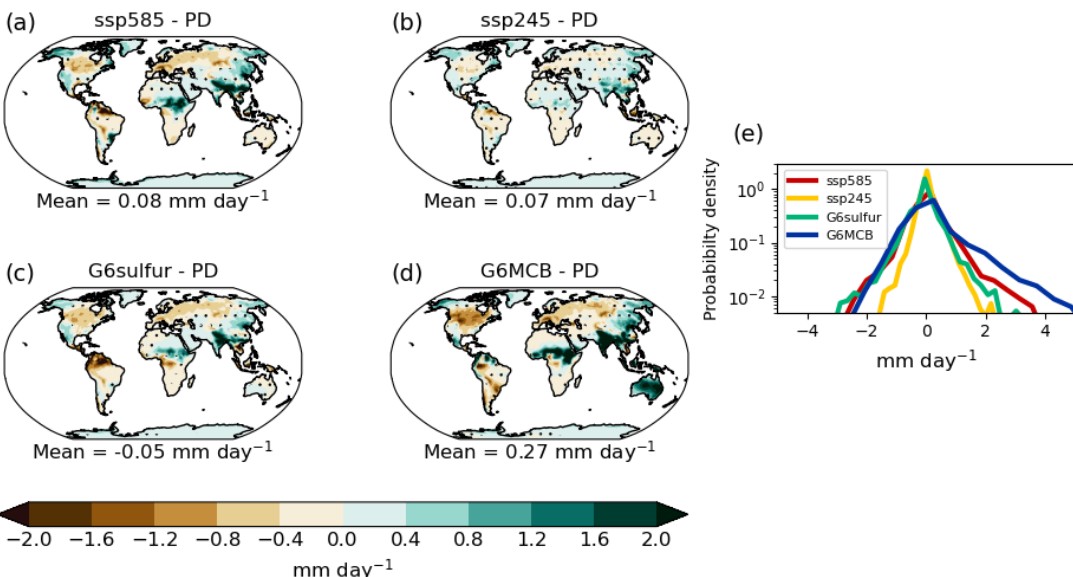

**Figure 11:** Same as Fig. 9 but for JJA land precipitation rate (mm day⁻¹).



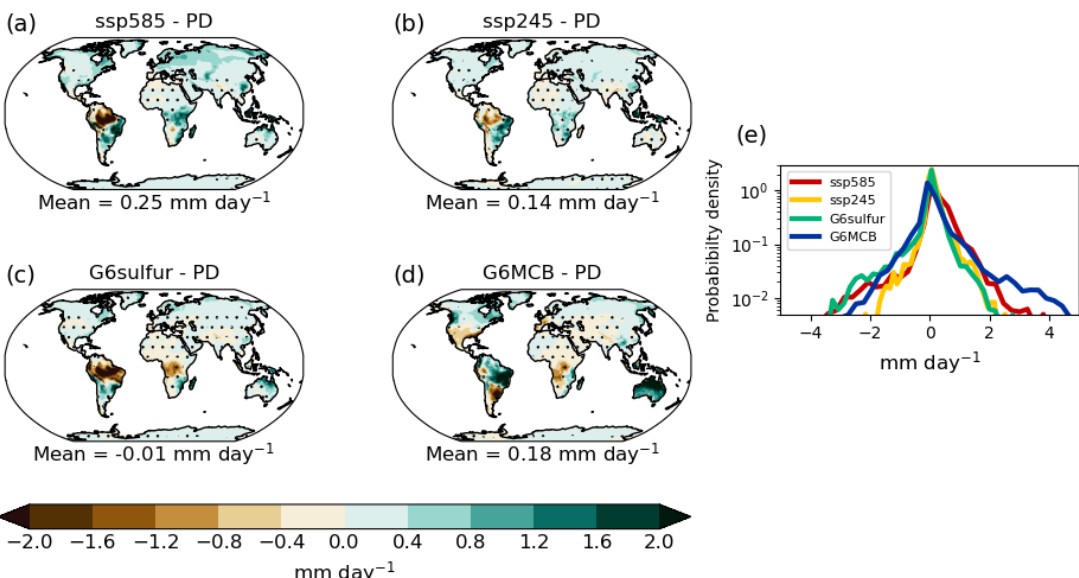

**Figure 12:** Same as Fig. 11 but for DJF.



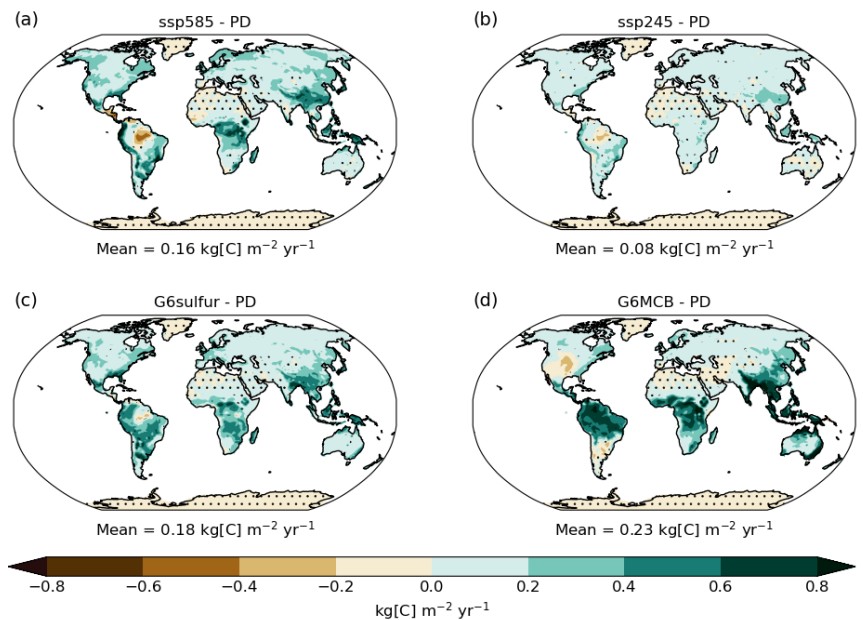

**Figure 13:** Change in annual-mean NPP (kg of carbon m$^{-2}$ yr$^{-1}$) for 2081-2100 compared with PD in (a) ssp585, (b) ssp245, (c) G6sulfur and (d) G6MCB. Stippled areas show where the differences are not significant at the 5% level in a two-tailed *t*-test.

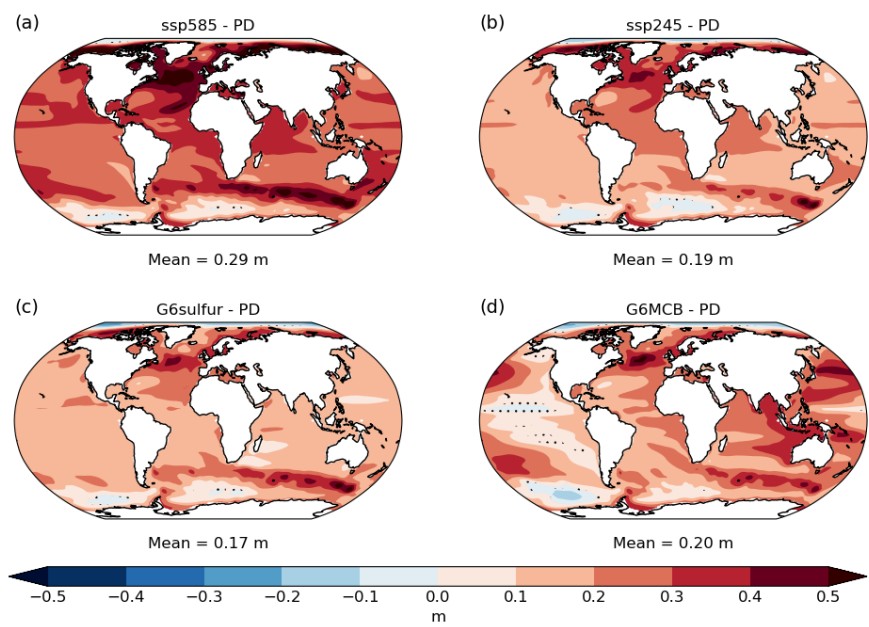

**Figure 14:** Change in sea-level for 2081-2100 compared with PD in (a) ssp585, (b) ssp245, (c) G6sulfur and (d) G6MCB. Stippled areas show where the differences are not significant at the 5% level in a two-tailed *t*-test.



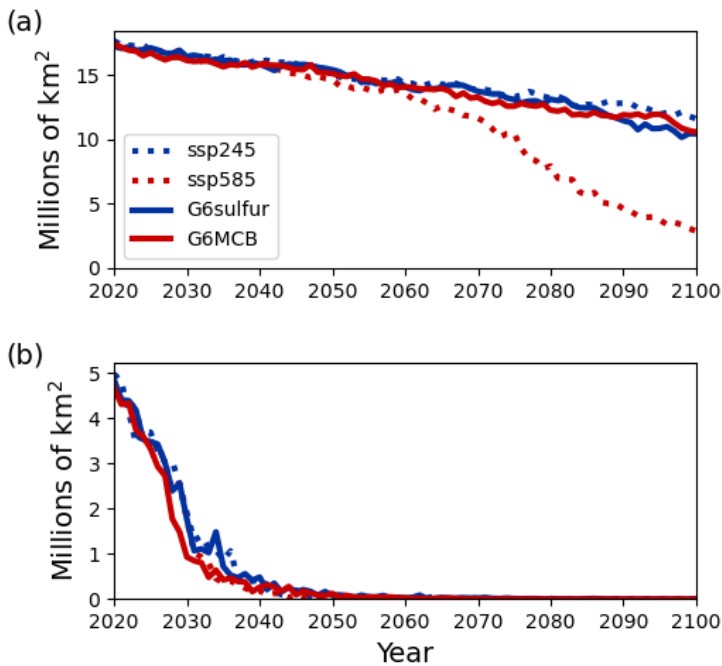

**Figure 15:** Arctic sea-ice area ($10^6$ km$^2$) for (a) March, showing the maximum sea-ice extent, and for (b) September, showing the minimum extent.

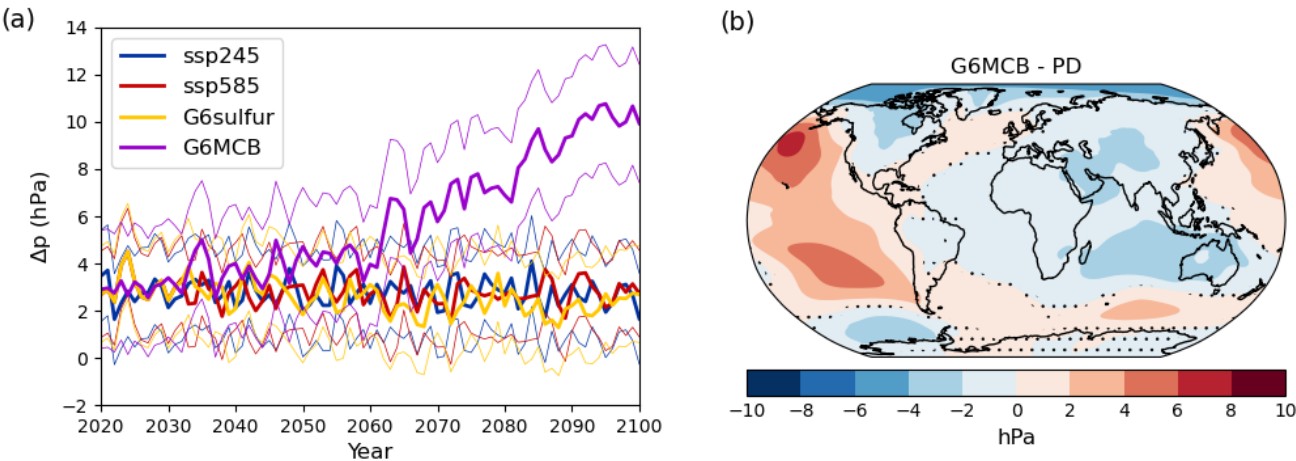


**Figure 16:** a) The SOI (hPa) derived as the simple difference in pressure between Tahiti and Darwin as a function of time for the simulations described in the text. The thick lines represent the ensemble mean and the thin lines the mean ± two standard deviations. b) The spatial distribution of the change in the pressure pattern (hPa) determined for 2081-2100 for G6MCB compared with present day (PD).

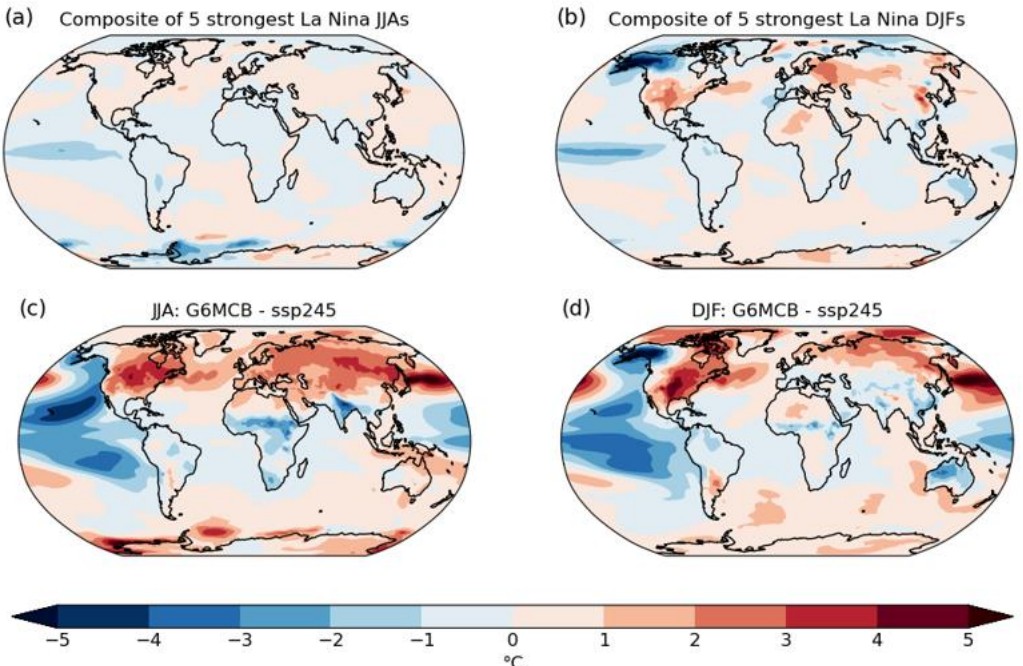

**Figure 17:** The patterns of near-surface temperature. Top row, perturbations for a) JJA and b) DJF diagnosed from the natural variability within the model as described in the text. Bottom row, perturbations for c) JJA and d) DJF diagnosed from the G6MCB-ssp245 simulations.

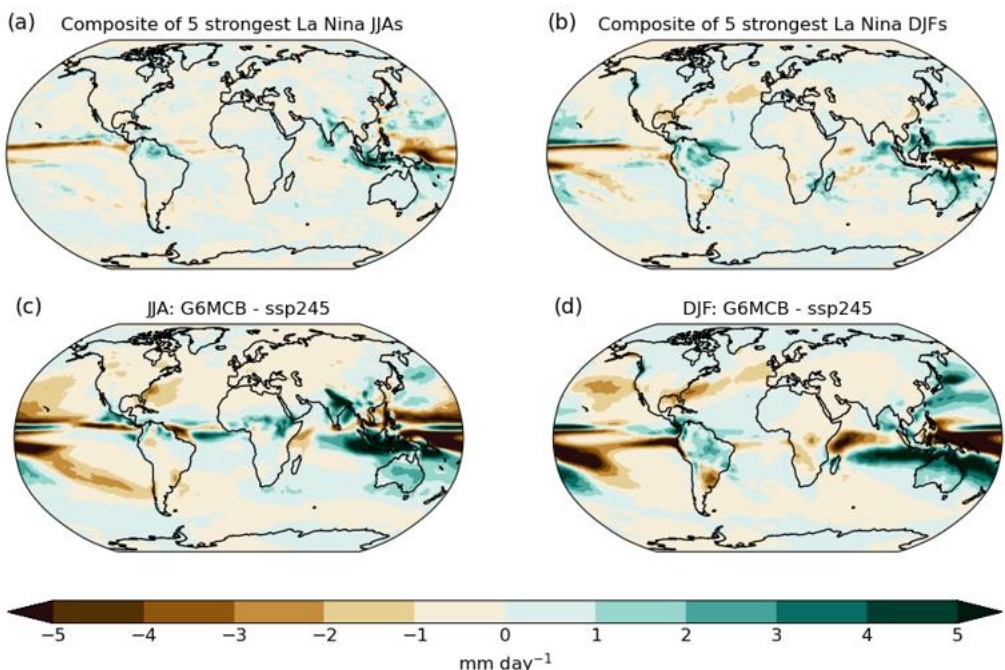

**Figure 18:** The patterns of precipitation. Top row, perturbations for a) JJA and b) DJF diagnosed from the natural variability within the model as described in the text. Bottom row, perturbations for c) JJA and d) DJF diagnosed from the G6MCB‑ssp245 simulations.