# Peer review of "Climate Intervention using marine cloud brightening (MCB) compared with stratospheric aerosol injection (SAI) in the UKESM1 climate model."

_EGUsphere, 2023_

## Author Comment (AC1)

We would like to thank the 3 reviewers for their comments on the manuscript. Each one of the reviewers is supportive of publication with relatively minor corrections and clarifications. We believe that, in responding to the reviewers' comments, we have significantly improved the clarity and comprehensiveness of the paper while maintaining its focus. We provide clarifications and corrections to address the reviewers' comments in red below.

The most significant amendment is a to provide some detail in the physical explanation of the results. To make the paper manageable and maintain the balance, we provide suitable references to previous work and a brief description of the oceanic circulation of the Pacific, which is dominated by the northern and southern gyres. While these physical explanations are far from comprehensive, the idea of cooling the eastern side of the Pacific does not seem like it will progress too much further owing to the detrimental impacts that we document. We believe that a comprehensive analysis that includes impacts on oceanic transport would be extremely prudent, once a more optimal deployment strategy is developed.

Where we have made corrections to figures, added figures (as suggested by the reviewers), or moved big chunks of text we do not use track changes in the revised document. This is simply because it becomes too difficult to follow in the track changes document if they are included.

Reviewer #1:

In this manuscript, the authors devise a new "G6MCB" experiment that parallels the GeoMIP-endorsed G6sulfur experiment but uses sea salt emissions in four regions of the Pacific rather than sulfur emissions in the tropical stratosphere to achieve a reduction in global mean surface temperatures to that of the SSP2-4.5 scenario from a background of SSP5-8.5. For lower levels of spraying with accumulation-mode particles, cloud brightening and the direct effect both produce a substantial cooling effect; at higher levels, cloud forcing saturates and even reverses with all of the cooling coming from the direct effect. The La Niña-like pattern induced by G6MCB produces marked differences in climate response from the warming and SAI scenarios to which it is compared. Perhaps the most striking result is that sea levels over the vulnerable western Pacific basin rise more in G6MCB than in ssp585 due to the La Niña-like dynamic adjustment to eastern Pacific cooling. The paper is well-written and executed and makes an interesting and important contribution to the literature surrounding MCB and SRM more broadly — it merits prompt publication following minor revisions (see specific points below). -Michael Diamond

We would like to thank Michael for his comprehensive review of the manuscript and are glad that, "The paper is well-written and executed and makes an interesting and important contribution to the literature surrounding MCB and SRM more broadly."

General point: At low-ish forcings (~1 W m-2), MCB dominates over MSB. The MSB findings may well be a result of pushing the system further than MCB can go. One conclusion may be that MCB is more feasible for a smaller scenario (e.g., sustaining historical peak aerosol cooling) but is infeasible for a more ambitious scenario (multiple degrees of cooling). It may be worth considering this point more explicitly in the discussion.

We already allude to this point in the discussion and conclusions, but we focus more on the fact that the ARG scheme may be pushed beyond the range of conditions that is was designed for. However, we take the point and modify the text from:-

"It is plausible that the ARG scheme may produce reasonable results when the injection rates of sea-salt are low, but that it becomes progressively less reasonable when the injection rates become very high."

"On the face of it, it might be concluded that MCB may be viable in delivering relatively modest radiative forcings of up to $-1Wm^{-2}$ for this particular injection strategy, but radiative forcings stronger than $-1Wm^{-2}$ may not be achievable through MCB. An alternative interpretation may be that the ARG scheme may produce reasonable results when the injection rates of sea-salt are low, but that it becomes progressively less reasonable when the injection rates become very high."

We also make more explicit statements to this effect in the abstract:-

"This deployment strategy appears capable of delivering a radiative forcing of up to $-1Wm^{-2}$ from MCB, but at higher injection rates, much of the radiative effect in G6MCB is found to derive from the direct interaction of the injected sea-salt aerosols with solar radiation, i.e. marine sky brightening (MSB)."

Specific points:

1. Why is the experiment named "G6MCB" instead of "G6sea-salt", which would be more parallel with the G4 naming system? I appreciate that this experiment is not officially within the GeoMIP umbrella, but it still seems like highlighting the sea-salt aspect may be more appropriate, especially as MSB dominates forcing by the end of the century.

This is a good point. As pointed out, this is a 'non-official' GeoMIP simulation. The various side-effects that we identify with this injection strategy mean that it is unlikely that this injection strategy will make it to an officially endorsed multi-model GeoMIP scenario. We chose to leave the 'G6sea-salt' nomenclature available for future official multi-model simulations (e.g. to parallel G4sea-salt).

2. Lines 16-18: This is true at high emissions rates; at lower emissions rates and with smaller particles the cloud effect dominated. This is probably worth clarifying, as the "lower" emission rates still produce forcings that may be policy relevant (e.g., targeting 1 W m-2 of cooling to maintain peak 20th century aerosol forcing).

Agreed, we modify the sentence to the following:- "This deployment strategy appears capable of delivering a radiative forcing of up to $-1Wm^{-2}$ from MCB, but at higher injection rates, much of the radiative effect in G6MCB is found to derive from the direct interaction of the injected sea-salt aerosols with solar radiation."

3. Section 3.1: It may be useful to mention or highlight the likely dependence on activation scheme here. This issue is dealt with nicely in the discussion, so perhaps you could just add an allusion to further information about the activation scheme question that will come later.

Good idea. We now include this sentence at the end of the section. Some of the implications and limitations of utilizing the Abdul-Razzak and Ghan (2000) activation scheme are highlighted in section 4. "Some of the implications and limitations of utilizing the Abdul-Razzak and Ghan (2000) activation scheme are highlighted in section 5."

4. Figure 3 caption: I don't understand the caveat about points that don't meet G6 standards. Could you clarify here or in the text?

We do already state this in the text: "As in the G6sulfur simulations, the goal of G6MCB was to reduce the global mean temperature from that of ssp585 to that of ssp245 to within ±0.2 °C for each decade from 2021-2100, and as with G6sulfur the sea-salt injection rates for each decade were determined by trial and error." However, we agree that the link could be better made to the figure caption, so we re-state this criteria explicitly in the figure caption.

5. Figure 4: Could you also plot the net forcing difference and perhaps the zero line? It's easy enough to eyeball but still could be useful for readers.

Agreed – amendments made in line with the reviewer comments.

6. Line 274: I agree with this point and it's important to make, but perhaps it should more specifically refer to generalizability when comparing MCB/MSB strategies with substantially different spatial patterns of forcing. The current (on-going) intercomparison seems to suggest that results are more generalizable when using a standardized protocol.

Agreed. We change the wording to:-

"This indicates a strong dependence of response on the chosen injection strategy and thus a lack of generalisability of results for MCB simulations with different injection strategies, indicating that standard emission protocols are required when reporting multi-model results."

7. La Niña section: A useful figure would be a comparison of G6MCB-SSP585 and the La Niña signal in UKESM1 from interannual variability (e.g., regression on detrended SOI or using some detrended Niño3.4-like index) in terms of variables like temperature, precipitation, sea level pressure, and perhaps circulation anomalies like surface winds.

In the submitted version of the manuscript, we infer La Nina-like conditions from the SOI and the pattern of the induced MSLP change and refer the reader to Trenberth and Shea (1987, TS87). The relevant correlation plot of MSLP of TS87 against the pressure in Darwin is shown below, together with the change in pressure patterns from our analysis (noting that the sign of the correlation in TS87 is reversed as they focus on El Nino, rather than La Nina conditions). The similarity is obvious – with two areas of strong low pressure in the observations in mid-latitudes over the eastern Pacific Ocean and an area of high pressure centred to the north west of Australia.

[Figure]

FIG. 1. Composite assessment of the correlations of annual mean sea level pressures with Darwin. In the Northern Hemisphere north of 15°N it originates from Trenberth and Paolino (1981) and in Australasia from Trenberth (1976). Elsewhere it originates from many sources, principally including Wright (1985).

However, we agree that there is probably some utility to discussing the La-Nina-like response in more detail focussing on the temperature and precipitation changes that are induced in natural variability in the model in the La-Nina state and the similarity of this state to changes induced in the G6MCB simulations.

Note that the results presented in the submitted manuscript (new Figs 9-12) show changes induced by the end of the 21st century (2081-2100) for G6MCB-Present Day. Thus they include a significant pattern associated with global warming. To isolate the La Nina-like response in the absence of global warming, we analyse G6MCB – ssp245. We also analyse the strongest five La Nina-like events from a century long pre-industrial simulation which has negligible temperature trend. A comparison of the patterns of response is now provided. This analysis has its own dedicated section.

8. Figure 15a: Put definition of the thin lines (two sigma?) in the figure caption.

This information is now included.

9. Lines 371-372: But isn't mean warming believed to be El Niño-like?

Yes – but this is just reporting what is found in the literature – the findings of Cai et al. (2015).

10. Line 377: Is "locking into La Niña" the right description? I'm interpreting the results here as showing a strong mean-state climate change pattern resembling La Niña, but in terms of interannual variability, do the frequency or intensity of El Niño and La Niña events change after adjusting for the changing mean state temperature/pressure?

Agreed – we now state: "A trend in the future mean climate into La Niña-like conditions….."

11. Lines 399-401: This makes sense, but given the inertia in the climate system, I'm not convinced it's correct. See, e.g., the results in MacMartin et al. (2022) that find a 10-year phase-out doesn't really differ from sudden termination in CESM2-WACCM.

MacMartin, D. G., Visioni, D., Kravitz, B., Richter, J. H., Felgenhauer, T., Lee, W. R., Morrow, D. R., Parson, E. A., and Sugiyama, M.: Scenarios for modeling solar radiation modification, Proc Natl Acad Sci U S A, 119, e2202230119, 10.1073/pnas.2202230119, 2022.

Point taken – it's probably more correct to simply state that MCB will be subject to the termination effect. We now state, "Note also that MCB will be susceptible to the termination effect if climate intervention is stopped abruptly (e.g., Jones et al., 2013, MacMartin et al., 2022) due to the short lifetime of MCB aerosols in the troposphere."

12. Data availability: I believe that the G6MCB data, minimally that needed to recreate the figures in the paper, need to be posted publicly to a data repository before publication, or "a detailed explanation of why" it is not available must be provided, to be compliant with the stated ACP data policy (https://www.atmospheric-chemistry-and-physics.net/policies/data_policy.html).

The G6sulfur data and ssp245 and ssp585 data is available from the GeoMIP and CMIP6 ESGF data nodes. We will provide this data for plotting G6MCB in a suitable repository before finalising the publication.
* * *
Reviewer #2

With the aim of advancing the understanding of climate intervention and assessing climate mitigation techniques, this study performs a set of simulations in the UKESM1 climate model, using sea salt aerosol injection (Marine Cloud Brightening, G6MCB) as compared to stratospheric aerosol injection (SAI, G6sulfur), to reduce global mean temperatures from SSP5-8.5 scenario to SSP2-4.5. The deployment strategy used in G6MCB injects sea-salt aerosol into four cloudy areas of the eastern Pacific. The authors find that much of the radiative effect in G6MCB is derived from the direct interaction of the injected sea-salt aerosols with solar radiation, rather than from aerosol-cloud interaction. The authors discuss the potential side effects of SAI and MCB, including overcooling of the tropics and residual warming of mid- and high latitudes, which are common for both SAI and MCB, and other side effects such as a strong La Nina like condition, that might depend on the choices of MCB emission scenario and the deployment strategy. I find this study very interesting and inspiring, and I believe it would certainly motivate future studies to better understand the complexity of MCB strategies and impact. I recommend acceptance but I do have some comments that I suggest the authors take into consideration.

We would like to thank the reviewer for their thorough review of the manuscript. We are pleased that the reviewer states, "I find this study very interesting and inspiring, and I believe it would certainly motivate future studies to better understand the complexity of MCB strategies and impact."

Main comments:

The authors recognize that the results from their G6MCB maybe an artifact of the model configuration (e.g., the choice of aerosol activation parameterization) that incorrectly represent water vapor competition at very high concentrations of small particles. This is a direct microphysical issue. Another potential issue would be related to the interaction between dynamics and microphysics in the model. E.g., how does the turbulence couple with microphysics in UKESM1? How would the cloud top entrainment change with cloud droplet size in UKESM1? Some discussion in this regard would be helpful.

We now introduce the aerosol indirect effect parameterisation within UKESM1 more thoroughly. We also agree that more discussion of the assumptions that are associated with modelling aerosol-cloud-interactions within the coarse resolution UKESM1 is

warranted. We think that this is an important point, so we include the following penultimate sentences in the concluding paragraph of the discussion, "A caveat with all studies reporting results from aerosol-cloud interactions within a coarse resolution Earth System Model, is that many of the microphysical processes such as cloud top cooling, subsidence, entrainment, detrainment, the representation of cloud base-updraft velocities etc. are not explicitly resolved or represented (e.g. Stevens and Feingold, 2009; Seifert et al., 2015; Haghighatnasab et al., 2022) which contributes to a significant uncertainty in results of global MCB studies. Large-scale effusive volcanic eruptions provide useful, but not perfect analogues for examining the representation of MCB within such coarse resolution models; the results reveal reasonable representation of the aerosol-induced observed perturbations to cloud droplet effective radius within coarse resolution climate models (e.g. Malavelle et al., 2017; Chen et al., 2022), but shortcomings in the representation of aerosol-induced perturbations to cloud fraction (e.g. Chen et al., 2022)."

The authors show some interesting side effects of MCB experiment including cooler tropics and warmer polar regions, and other regional changes in temperature and precipitation, yet the authors provide no physical explanation/hypothesis regarding the potential mechanisms. It would potentially be more compelling if the authors could connect MCB with large scale circulation change.

In the submitted version of the paper, we had 15 figures. This has increased to 18 owing to request for additional information on the baseline aerosol distribution and analysis of the La Niña-like nature of the MCB-induced climate change which we diagnose somewhat differently to remove the confounding impacts of global warming that are included in new Figs 9-12 (see response to reviewer #1). We now provide a brief synopsis of the mechanisms behind the response behind both SAI and MCB response. We now include the following text in the discussion:-

"Multi-model GeoMIP studies have documented that reducing the solar constant by a fixed fraction reduces downward shortwave flux by a greater amount in the tropics than at the poles and will have no impact at all in wintertime for polar regions where there is no solar irradiance (Kravitz et al., 2013). In addition, the fact that UKESM1 exhibits a strong tropical pipe that isolates the tropical stratosphere from the mid-latitudes inhibits poleward transport of aerosols, resulting in an aerosol optical depth that is much greater in tropical regions than over the poles (e.g. Figure 6a and Visioni et al., 2023). Thus, G6sulfur shows the expected maximum zonal mean residual warming for 2081-2100 between 60-90 °N which has been evident in GeoMIP simulations which inject aerosol at Equatorial latitudes (e.g., Kravitz et al., 2013a, 2015)."

"For MCB, in the northern hemisphere, much of the cooling impact from MCB is confined to the low-latitude and eastern Pacific, accompanied by warming in the Kuroshio and North Pacific Current region (Fig 17). This Pacific Decadal Oscillation (PDO)-like pattern of SST change, like the PDO itself (Newman et al., 2016), appears to arise from a combination of multiple oceanographic and atmospheric processes. Enhancement of the high pressure systems sitting above the subtropical North and South Pacific in response to MCB (Fig 16b) will impact the ocean in a number of ways. (1) Increased equatorward windspeeds along the west coasts of North and South America, will increase Ekman transport and upwelling of cool water along those coasts, supressing SSTs towards the east of the basin. (2) Increased anticyclonic movement of air above the subtropical gyres will result in increased geostrophic flow within the gyres, evidenced by positive sea surface height anomalies over the gyres (Fig 14d). With a strengthening of the subtropical gyre circulation there will be an increase in southward then westward transport of cool waters on the

eastern side of the basin, and an increased northward transport of warm water in the western side of the basin. (3) Strengthening of the subtropical gyres will result in increased Sverdrup transport equatorward across the gyres, balanced by an enhancement of the western boundary currents (Vallis et al., 2017), in the case of the North Pacific, the Kuroshio current. Strengthening of the Kuroshio current will transport more warm equatorial water, more quickly, to the inter-gyre boundary region, where the secondary maximum in SSTs is seen (Fig. 17). Thus, while the overcooling in the tropics in SAI simulations is linked to changes in the surface irradiance, for MCB the overcooling in tropical regions in this study appears to be influenced by the ocean circulation."

We also add in the conclusions:-

"The very inhomogeneous forcing of MCB as applied in this scenario, appears to induce specific changes in the oceanic circulation in the Pacific sub-tropical gyres that transport the MCB-induced SST perturbations equator-wards and westwards. While SAI has been examined for the most part by atmospheric scientists, for MCB it appears essential to include more detailed analyses by oceanographers to better fully understand and quantify any potential impacts."

Are the areas of injection regions identical in both hemispheres?

We do already explicitly state the injection areas: "Sea-salt for climate intervention was emitted concurrently and at the same rate in four ocean regions designated NP (north Pacific: 30°-50° N, 170°-240° E), NEP (north-east Pacific: 0°-30° N, 210°-250° E), SEP (south-east Pacific: 0°-30° S, 250°-290° E) and SP (south Pacific: 30°-50° S, 190°-270° E) as shown in Fig. 2. Within the latitude-longitude ranges indicated, only those model grid-cells which were 100% ocean were used for sea-salt injection."

By design, the areas of injection are very similar in size. For the Northern Hemisphere, the area is N. 26.09 million $km^2$, while for the southern hemisphere the area is 27.25 million $km^2$.

The NEP and SEP regions have the same lat-lon extents but a much bigger bite is taken out of SEP by land than NEP (see Fig. 2). NP and SP have the same N-S extents but SP is 10 degrees wider in the E-W direction. In hemispheric terms this slightly overcompensates for the differing amounts of land intruding into SEP and NEP, giving the numbers above.

This information is now given.

Does the quantity of injected aerosols exhibit hemispherical symmetry? Or are there actually more injected aerosols in one hemisphere?

The amount of aerosols injected in each hemisphere is roughly equivalent (within 4.5%). This information is now given.

The authors talk about potential asymmetry in near-surface air temperature in Fig. 7. They claim the asymmetry might depend on the deployment strategy. I wonder if the authors can make a similar plot for albedo. I wonder how the results are related to the idea of all-sky albedo symmetry where the cloud adjustment would potentially balance the aerosol hemispheric asymmetry.

This is an interesting question. However, we have now provided more detail on the MCB aerosol injection strategy – by design it is essentially equivalent between the two hemispheres with the specific objectives of reducing any influence on the position of the ITCZ (e.g. Haywood et al., 2013). As noted in the text, it appears any forcing (e.g. stratospheric gradients in AOD, changes in reflectivity of tropospheric cloud, or surface reflectance) that induce strong cross-equatorial temperature gradients induce shifts in the ITCZ (e.g. Haywood et al., 2016). As we have already provided plots of the change in the top of atmosphere radiative impacts of the MCB (e.g. Fig. 6), we chose not to follow this suggestion.

Specific comments:

More description of model setup is needed.

The results in Fig. 2 are from 10-year simulations which differ from the other 80-year simulations in the manuscript, but this distinction is not clarified until Section 3. I think all the descriptions of model setup should go to Section 2. To enhance clarity, it would be beneficial if the authors could make a table summarizing the simulation design used in this study. This table could include information such as initial conditions, total simulation time (including spin-up), and the specific time frame used for analysis. The current description appears too simple. E.g., there is no specification as to when the 15-year simulation starts. Could the authors elaborate on their rationale for selecting a 15-year duration, rather than a longer one? I am curious about the potential sensitivity of the results to the start time and duration of the simulations.

We agree that there was a partial mix between results and experimental set-up. We have taken the reviewer's suggestion and included the preliminary experiments to determine the optimal injection size in the experimental set-up. This makes things much easier to understand as we move from G6sulfur, through the preliminary MCB simulations that are needed to optimise G6MCB through to the G6MCB simulations themselves. The new restructuring should be easier to follow and we don't think that a Table is necessary as the simulations (ssp245, ssp585, piControl, G6sulfur) are standard CMIP6 simulations of GeoMIP simulations that have been documented elsewhere.

Lines 153-155: The description should go to Section 2.

Agreed and modified.

Line 159: Is natural sea-salt emission included in the baseline experiment (ssp245)?

Yes – this is clarified.

Lines 163-164: What is the typical particle size for G6sulfur? What is the typical aerosol lifetime near the surface and in the stratosphere? More information is needed here.

The G6sulfur simulations have already featured in a number of publications (e.g. Jones et al., 2021, 2022, Visioni et al., 2021). We already state this, "Results from UKESM1's G6sulfur experiment have been documented in previous studies, e.g., Jones et al. (2021) and Visioni et al. (2021)." In our view, it seems unnecessary to present results again ……

Line 170-174: Could the authors explain why temperature respond linearly to aerosol injection over a limited range, but non-linearly over a wider range?

We've added a few words. The stratospheric sulfate aerosol size increases owing to the deposition of sulfur dioxide onto pre-existing particles, but stress that this was the finding from the study of Niemeier and Timmreck (2015), "owing to the increase in particle size which decreases the scattering efficiency per unit mass at solar wavelengths, and also increases the aerosol sedimentation rate"

Line 203: Could the authors elaborate on the dynamical feedbacks that lead to positive CRE_SW?

We realise that this was rather brief. To bolster the analysis we add the following text in the discussion and conclusions;-

"In our study, while the microphysical impacts of clouds are evident at more modest injection rates (Fig 6a), the dynamical response of clouds becomes increasingly important as the injection rates increase (Fig 6b). Robust observational correlations between cloud fraction and SSTs have been developed on a regional basis from observations (e.g. Warren et al., 2007; Eastman et al., 2011) which reveal strong negative correlations between SSTs and clouds (i.e colder SSTs lead to more clouds) in regions of upwelling over the eastern pacific, which transition to strong positive correlations (i.e. colder SSTs lead to less clouds) in the central Pacific. In our simulations, the strong local cooling that is induced over the eastern Pacific by the MCB is advected equatorward and then westward, leading to an SST-related reduction in cloud fraction over the central and western Pacific. These model results are therefore in line with observations that relate SSTs to cloud fraction (Eastman et al., 2011) and also with observations of the response of clouds to La Niña-like conditions which are discussed (Park and Leovy, 2004) in more detail later."

For the reviewer's convenience, here is the relevant figure of correlations (black = negative, big = strong) from Eastman et al (2011).

[Figure]

Eastman, R., Warren, S. G., & Hahn, C. J. (2011). Variations in cloud cover and cloud types over the ocean from surface observations, 1954–2008. Journal of Climate, 24(22), 5914-5934.

Line 229-235: Could the authors explain why G6MCB also leads to cooler tropics and warmer polar regions?

We now include a dedicated paragraph on this in the discussion and conclusions. This is linked to the strengthening of the Pacific sub-tropical gyres in both hemispheres.

Line 237-257: Could the authors provide a physical explanation/hypothesis for the disparity observed between G6MCB and G6sulfur, as shown in Figs. 9 and 10?

See above.

Line 259: Please define net primary productivity.

OK.
* * *
Reviewer #3

The manuscript outlines the results from a pair of geoengineering experiments performed using the UKESM1 model. I find that the subject matter to be appropriate for the ACP journal and the results are scientifically interesting. Overall, the results are well documented, and appropriate connections with existing literature are made. However, I find that the following two areas can benefit from a better interpretation or more analysis.

First, the authors appear to suggest in the discussions and conclusions section that the shift from cooling to warming aerosol indirect effect response is due to a strong competition of water vapor by a very large increase in aerosols. However, Figure 5 shows that the large positive response do not occur where the aerosols are injected but in other regions. A clarification of what is behind the positive cloudy-sky effect in Fig 5b is needed. If the authors want to make the case that the positive response is indeed due to a decrease in relative humidity due to water competition, more evidence is necessary.

Second, I found the interpretation that the precipitation and sea-level rise pattern changes are connected to a La Nina-like SST pattern to be quite interesting and compelling. However, this connection was only mentioned in the discussions and lacked the analysis of the other sections. Assuming that the UKESM1 model produces El Nino and La Nina events, I find that an even stronger case can be made if the authors showed the precipitation and sea-level difference patterns composited on La Nina patterns. Given the asymmetric nature of the forcing induced by the MCB strategy, then one might then infer that these precipitation pattern changes would likely be robust across different models.

We would like to thank Reviewer #3 for their constructive comments. We are pleased that the reviewer states, "Overall, the results are well documented, and appropriate connections with existing literature are made."

The two general comments are dealt with below:-

1) We agree that the argument was not presented correctly. We focussed too much on what other had found rather than what our results showed. We have rectified this by including a dedicated paragraph in the discussion and conclusions (see response to Reviewer #2).

2) We now provide an additional analysis where we remove the impacts of global warming from the analyses (by examining G6MCB - ssp245 where the global mean temperature changes are very similar). We also do as the reviewer suggests by examining the model La Niña responses from a century long steady-state simulation where we diagnose the 5 strongest La Niña events for DJF and JJA.

Because of this extension to the analysis we choose to include a new section 4 "How La Niña-like is the response in G6MCB?" This includes Figure 16a and b (pressure difference patterns and the SOI trend) and new figures for temperature (Fig 17) and precipitation (Fig 18). Please also refer to the response to point #7 of Reviewer #1,

Specific comments

L159: I assume this is 10% of the global estimate of natural sea-salt emission rate. It would be useful to know what fraction of the ocean is covered by these injection sites – although it might be 10% of the global mean, it might be doubling (or more) the sea-salt mass emissions in these regions.

Yes – we do explicitly state that this is the global sea-salt emission rate: "10% of estimates the observed natural global sea-salt emission rate, although the latter has a large degree of uncertainty (Lewis and Schwartz, 2004)". From the ratios of the aerosol optical depth that are shown in the lower panels of figure 7, it is immediately obvious that the aerosol emissions must be significantly greater than the natural emissions.

Figure 5: It would be helpful to see the map of the mean-state shortwave cloud radiative effect in the baseline (maybe the ssp245) to see where the cloud changes are occurring relative to the stratocumulus cloud decks and how large their changes are relative to the background cloud radiative effect.

We don't think that this is particularly useful. Please keep in mind that the plots are of the SW cloud radiative effect. It's very easy to misinterpret the SW CRE that we show as the net CRE (i.e. the sum of the SW and LW). If we were to present the figure below that casual reader would get the erroneous impression that the stratocumulus regions that we are targeting contribute little to cooling via the negative cloud radiative forcing.

[Figure]

(a)  SW CRE for PD

Mean = -43.68 W m$^{-2}$

-100  -80  -60  -40  -20   0
W m$^{-2}$

L202: I am guessing this increase in the cloudy-sky effect is due to remote impacts due to atmospheric dynamics response or more La Nina like conditions with the sea salt injection. If this is the case, this point should be made clearer in the conclusions. At least, it should be noted in the conclusions that the positive cloud radiative effect response is occurring away from the sea-salt injection sites.

We agree with the reviewer's assessment. We have revised the text in line with the reviewer comments with substantially more focus both on the modelled remote response of clouds (that is in line with observed correlations) and a dedicated section on the La Nina-like response.

L204-206: Are the locations of the more positive cloudy sky effect consistent across the simulations? It is suggested that cloud droplet activation scheme might be a reason for the cloud response, but seeing that the positive CRE_SW occurs in the Tropical West Pacific and South Pacific Convergence Zone, away from the injection sites, it seems that the change in SST patterns might be playing a role here.

Agreed – this is now clarified.

Figure 6: As in Figure 5, having a map of the PD AOD will help see how the changes in AOD compare with the background.

We chose to add the PD AOD in the description of the model in a new Figure 1. We feel that the specific inclusion of the lower two panels in Figure 6 (now 7) provides exactly what the reviewer is asking for – how changes compare to the background. For example the lower panels show increases in AOD on up to a factor of 10 for SAI and up to a factor of 20 for MCB.

L315-322: This interpretation and discussion of the negative aerosol indirect effect is misleading, since the forcing change (from negative to positive) does not occur over the region of injection, but on the other side of the Pacific Ocean basin. Instead of an aerosol indirect effect saturation or a swap of signs, it seems that this change in global ACI response is due to a change in the SST pattern that occurs from strongly cooling the Eastern Pacific, relative to the Western Pacific.

Agreed. This was misleading. A new paragraph has been added to the discussion and conclusions.

L348-353: It is suggested that the La Nina-like conditions that set up in the model are the reason for the changes in the precipitation and sea-level rise response. If that is the case, we should be able to see these patterns show up if the La Nina cases in the model are composited from the UKESM1 model. Such a pattern will lend more support for this argument and also distinguish how much of the precipitation pattern and cloud changes are due to indirect SST influences or direct aerosol increases.

We have performed an analysis, which does tend to support the La Nina-like nature of the response. Of course, its much bigger in the MCB simulations than in the natural variations in the model. But this is an important point in itself as there are many impacts of La Nina (e.g. fisheries), but the far greater magnitude of the MCB-induced La Nina-like perturbations means that the system would be pushed outside its natural bounds.

Technical corrections

Figure 15: What do the thin lines represent? Please indicate in the captions.

Added.